# Efficient Verified Machine Unlearning for Distillation

**Yijun Quan, Zushu Li, and Giovanni Montana**
Warwick Manufacturing Group
University of Warwick
CV4 7AL
{yijun.quan, z.li.19, g.montana}@warwick.ac.uk

## Abstract

Growing data privacy demands, driven by regulations like GDPR and CCPA, require machine unlearning methods capable of swiftly removing the influence of specific training points. Although verified approaches like SISA, using data slicing and checkpointing, achieve efficient unlearning for single models by reverting to intermediate states, these methods struggle in teacher-student knowledge distillation settings. Unlearning in the teacher typically forces costly, complete student retraining due to pervasive information propagation during distillation. Our primary contribution is PURGE (Partitioned Unlearning with Retraining Guarantee for Ensembles), a novel framework integrating verified unlearning with distillation. We introduce constituent mapping and an incremental multi-teacher strategy that partitions the distillation process, confines each teacher constituent model's impact to distinct student data subsets, and crucially maintains data isolation. The PURGE framework substantially reduces retraining overhead—requiring only partial student updates—when teacher-side unlearning occurs. We provide both theoretical analysis, quantifying significant speed-ups in the unlearning process, and empirical validation on multiple datasets, demonstrating that PURGE achieves these efficiency gains while maintaining student accuracy comparable to standard baselines.

## 1 Introduction

Growing data privacy demands, driven by a global wave of data privacy regulations exemplified by the General Data Protection Regulation (GDPR) and California Consumer Privacy Act (CCPA), grant users the right to retract their data from machine learning models. Fulfilling these rights is critical, as models can potentially memorize training data [10, 4], necessitating alterations to trained models upon data retraction requests. However, naively retraining modern deep neural networks, which can contain billions of parameters, from scratch after each data removal is computationally prohibitive and economically unviable, especially given the potential frequency of such requests. This necessitates efficient machine unlearning techniques. Furthermore, many applications demand *verified* unlearning methods that formally guarantee the complete removal of data influence, a property often lacking in approximate or model-specific approaches which may only offer heuristic or probabilistic removal guarantees[25, 32, 19]. The challenge lies in developing methods that are both computationally efficient and provably effective in removing data influence.

The Sharded, Isolated, Sliced, and Aggregated (SISA) framework [1] offers a prominent solution for achieving verified, model-agnostic unlearning. SISA partitions the training data into isolated shards and trains an ensemble of constituent models, where each model learns exclusively from its assigned shard. Training proceeds incrementally on data 'slices' within each shard, with intermediate model checkpoints saved after processing each slice. This inherent isolation ensures that unlearning a data point typically requires only reverting the single affected constituent model to a relevant prior state

39th Conference on Neural Information Processing Systems (NeurIPS 2025).

and partially retraining it on a small fraction of data. This mechanism provides exact unlearning guarantees (matching the model distribution as if trained without the removed data) while offering significant efficiency gains over full retraining in standard single-model scenarios.

While SISA is powerful for individual models, adapting verified unlearning effectively to more complex learning paradigms like Knowledge Distillation (KD) poses unique, previously unaddressed challenges. KD is crucial in modern machine learning, enabling the deployment of state-of-the-art capabilities by transferring knowledge from large, computationally intensive 'teacher' models (often trained on vast datasets) to smaller, more efficient 'student' models suitable for resource-constrained environments or low-latency applications [15, 5, 26]. During this distillation process, however, information about the teacher's training data can leak and propagate pervasively throughout the student network [24]. Consequently, even if SISA's partitioning and checkpointing are applied independently to both teacher and student ensembles, unlearning data from the teacher side forces costly, complete retraining of the *entire* student network. This occurs because all student constituent models are exposed to influence derived from the original, complete teacher ensemble during initial training, fundamentally breaking the data isolation necessary for efficient unlearning when the teacher model is updated. This critical issue negates SISA's efficiency benefits within the coupled teacher-student system, hindering the practical application of verified unlearning in common KD pipelines.

To address this critical gap, we propose PURGE (Partitioned Unlearning with Retraining Guarantee for Ensembles), a novel framework specifically designed for efficient and verified unlearning within the KD paradigm, focusing particularly on the challenge of teacher-side updates. The PURGE framework integrates SISA with KD by introducing constituent mapping, where each teacher constituent model's influence is restricted to a dedicated subset of student constituent models, and an incremental multi-teacher strategy for managing the distillation flow within student shards. This structure crucially maintains data isolation during the distillation phase itself, preventing the cross-model information propagation that plagues naive SISA applications in KD. PURGE enables efficient student unlearning: only a small, targeted fraction of the student network needs retraining when teacher data is removed, thereby restoring the efficiency promise of SISA for the entire system. Evaluated on both image classification and sentiment analysis tasks using public datasets, including MNIST[8], SVHN[12], CIFAR-100[20] and SST5[28], our proposed PURGE delivers significant speed-ups over SISA while preserving model performance across various conditions.

Our key contributions are: (1) the first framework, to our knowledge, providing verified unlearning specifically tailored for distillation scenarios involving teacher updates; (2) the novel mapping and incremental multi-teacher mechanism designed to preserve data isolation during distillation; (3) theoretical analysis quantifying the significant retraining speedups achieved by our method; and (4) empirical validation on multiple datasets demonstrating substantial practical efficiency gains without sacrificing student predictive performance compared to relevant baselines.

## 2    Related Work

Machine unlearning aims to efficiently remove the influence of specific data points from trained models, driven largely by data privacy regulations and the need to correct data errors. Broadly, approaches can be categorized into (1) approximate unlearning and (2) exact (or verified) unlearning.

*Approximate unlearning* often rely on heuristics such as proposing unlearning as learning with negative updates [3], using influence functions or Newton updates [13], model scrubbing [11], or leveraging connections to differential privacy [27] to estimate and counteract the contribution of data points to be removed. While potentially faster or applicable in specific settings (e.g., convex models [13, 27]), these methods typically lack formal guarantees regarding the complete removal of data influence for general deep learning models, and therefore may fall short of meeting the strict requirements of certain data privacy regulations.

*Verified unlearning* methods, in contrast, aim to produce a model state identical in distribution to one trained without the removed data. The Sharded, Isolated, Sliced, and Aggregated (SISA) framework [1] is a leading approach in this category. As outlined in Section 1, SISA achieves verified, model-agnostic unlearning through data partitioning (sharding), incremental training on data slices, and checkpointing, enabling efficient retraining of only isolated constituent models when data is removed.

While powerful, the architectural assumptions of SISA mean its effective application often requires adaptation for specific machine learning paradigms beyond standard supervised learning on independent data. For instance, significant research has explored adapting SISA for Federated Learning (FL), addressing challenges related to decentralized data and communication constraints [29]. Similarly, applying SISA to Graph Neural Networks (GNNs) necessitates handling graph structure dependencies [6], and adaptations exist for non-differentiable models like random forests where partitioning applies to the model structure itself [2]. Further demonstrating this need, Kumar et al. adapted SISA principles for large NLP models by retraining only lightweight adapter layers within shards to manage the high computational and memory costs [21]. This pattern highlights that achieving efficient verified unlearning often demands tailored solutions that respect the constraints and information flow of the target learning paradigm.

Knowledge Distillation (KD) presents another such paradigm with unique challenges for unlearning. As discussed, KD transfers knowledge from a teacher to a student model, a process crucial for model compression and deployment. However, this knowledge transfer intrinsically creates dependencies: information about the teacher's training data can leak to the student [24], complicating unlearning. Research addressing unlearning specifically within KD is sparse and has focused on approximate methods or student-side updates. For instance, SCRUB [22] trains a new student model to selectively "disobey" the original teacher on data intended for forgetting, offering no formal unlearning guarantees and leaving the original teacher model unchanged. RKLD [30] uses a supposedly "clean" reference teacher model to guide the original model (acting as a student) via reverse KL divergence to forget specific information, again lacking formal verification and relying on the availability of a suitable reference model. Other related works also utilize KD or fine-tuning primarily for approximate unlearning speedups on the student side [18, 17]. To our knowledge, no existing method provides efficient, verified unlearning directly applicable to the teacher model within a KD pipeline using SISA's partitioning principles. Thus, the most direct verified approach, applying SISA independently to both ensembles, fails for teacher-side unlearning, as information propagation during initial distillation forces costly full retraining of the student network whenever the teacher model is updated.

Our work, PURGE, directly addresses this gap by proposing the first SISA-based framework, to our knowledge, specifically designed for efficient and verified unlearning in KD, particularly handling teacher-side updates. Unlike approximate methods [3, 13, 11, 27, 22, 30, 18, 17], PURGE provides exact unlearning guarantees inherited from SISA. Critically, unlike the naive application of SISA to KD, and distinct from prior KD-unlearning techniques that focus on student retraining or require reference models, PURGE employs constituent mapping and an incremental multi-teacher distillation strategy (detailed in Section 3) to maintain data isolation during the distillation process while preserving student model performance. This structural modification prevents the information propagation problem and enables efficient, partial retraining of the student network when the teacher unlearns, making verified unlearning practical in teacher-student pipelines.

## 3   Methodology

The effectiveness of the SISA framework comes from the data isolation introduced by making each constituent model only trained on its corresponding shard without access to data in other shards. Such isolation is broken when the teacher network is used to train the student network in the standard distillation setup described previously. By using the teacher network as an entirety (providing a single supervisory signal derived from the full teacher ensemble), every constituent model of the student network gains indirect access to information influenced by all the data used to train the teacher network. As a result, any change in the upstream teacher network, such as unlearning a data point, necessitates updates that propagate to every downstream student constituent model, mandating the full, costly retraining of the student network. Clearly, the key to addressing this problem and restoring unlearning efficiency is to maintain isolation between the influence of different teacher data shards *within* the student network's training process as well. Thus, we propose a student-teacher constituent mapping strategy designed to isolate the influence of data associated with specific teacher constituent models to only a limited subset of student constituent models.

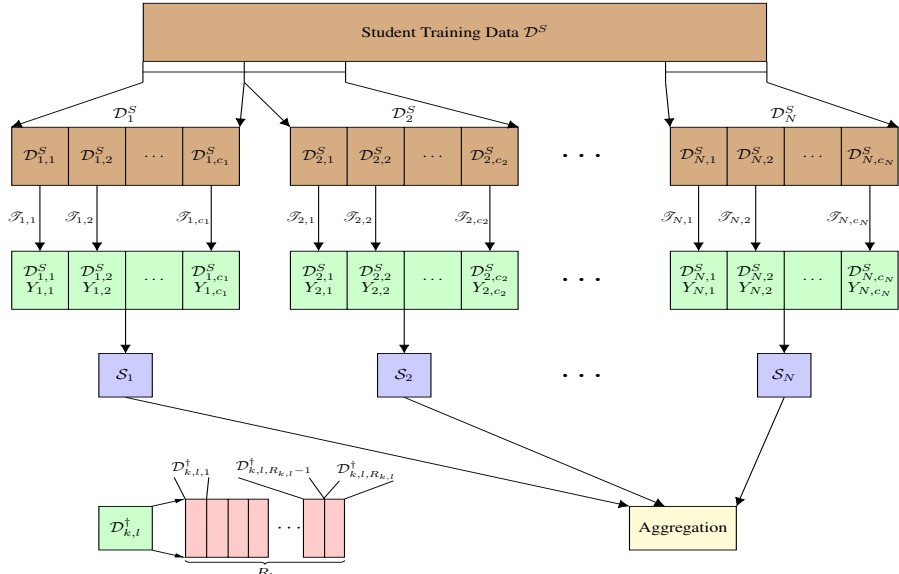

Figure 1: Overview of the proposed framework (PURGE) integrating SISA with knowledge distillation for efficient, verified unlearning. The structure maintains data isolation during distillation, enabling efficient student retraining upon teacher updates. Key steps: (1) Sharding: Student data $\mathcal{D}^S$ is partitioned into $N$ shards ($\mathcal{D}_k^S$), each assigned to a student constituent model ($\mathcal{S}_k$). (2) Mapping: Each $\mathcal{S}_k$ is mapped to a distinct teacher ensemble $\mathcal{T}_k = \{\mathcal{T}_{k,1}, ..., \mathcal{T}_{k,c_k}\}$. (3) Incremental Distillation: The student shard $\mathcal{D}_k^S$ is processed in $c_k$ sequential chunks ($\mathcal{D}_{k,l}^S$). Crucially, soft labels ($Y_{k,l}$) for chunk $l$ are generated only by an incrementally growing teacher subensemble $\mathcal{T}_{k,l} = \cup_{i \in [l]} \mathcal{T}_{k,i}$, limiting information propagation from the full teacher ensemble. (4) SISA Slicing & Training: Each resulting data-label chunk $\mathcal{D}_{k,l}^\dagger = [\mathcal{D}_{k,l}^S, Y_{k,l}]$ is further divided into $R_l$ slices ($\mathcal{D}_{k,l,j}^\dagger$). $\mathcal{S}_k$ trains incrementally on these slices using standard SISA checkpointing. (5) Aggregation: Final predictions are aggregated from all trained student constituent models $\{\mathcal{S}_k\}$. This design ensures that unlearning affecting teacher $\mathcal{T}_{k,i}$ only requires partial retraining of the corresponding student $\mathcal{S}_k$.

## 3.1 The PURGE framework

The PURGE framework maintains the data isolation required for efficient unlearning within the distillation process by implementing a strategy based on mapping specific teacher constituent models to specific student constituent models. This prevents the problematic information propagation identified in Section 2. Figure 1 illustrates the PURGE framework.

We consider a setup with $M$ teacher constituent models, $\{\mathcal{T}_1, \mathcal{T}_2, \ldots, \mathcal{T}_M\}$, and $N$ student constituent models, $\{\mathcal{S}_1, \mathcal{S}_2, \ldots, \mathcal{S}_N\}$, where $M$ and $N$ are not necessarily equal. Our core idea within PURGE is to partition the set of teacher constituent models into $N$ disjoint ensembles, $\mathcal{T}_1, \ldots, \mathcal{T}_N$, such that each student constituent model $\mathcal{S}_k$ learns exclusively from the teachers in its assigned ensemble $\mathcal{T}_k$. Let $\mathcal{T}_k = \{\mathcal{T}_{k,1}, \mathcal{T}_{k,2}, \ldots, \mathcal{T}_{k,c_k}\}$ be the ensemble assigned to $\mathcal{S}_k$, containing $c_k$ teacher constituent models (this number can vary per student). Crucially, each teacher constituent model $\mathcal{T}_m$ belongs to exactly one student's teacher ensemble: $\cap_{k \in [N]} \mathcal{T}_k = \emptyset$ and $\cup_{k \in [N]} \mathcal{T}_k = \{\mathcal{T}_1, \ldots, \mathcal{T}_M\}$. This strict mapping ensures that if unlearning affects a single teacher constituent model $\mathcal{T}_{k,i}$, only the corresponding student constituent model $\mathcal{S}_k$ will potentially need retraining.

To implement the distillation under this mapping, we first follow SISA by dividing the student dataset $\mathcal{D}^S$ into $N$ disjoint shards $\{\mathcal{D}_1^S, \ldots, \mathcal{D}_N^S\}$ ($\cup_{k \in [N]} \mathcal{D}_k^S = \mathcal{D}^S$, $\cap_{k \in [N]} \mathcal{D}_k^S = \emptyset$), where shard $\mathcal{D}_k^S$ is used for training $\mathcal{S}_k$. The PURGE framework then introduces one further level of partitioning (Chunking) specific to its design and applies SISA's Slicing methodology within these chunks:

1. **Chunking:** Each student data shard $\mathcal{D}_k^S$ is further divided into $c_k$ ordered, disjoint data chunks $\{\mathcal{D}_{k,1}^S, \ldots, \mathcal{D}_{k,c_k}^S\}$, where $c_k$ is the number of teacher constituent models mapped to $\mathcal{S}_k$.

2. **Incremental Multi-Teacher Distillation:** Soft labels for training $\mathcal{S}_k$ are generated chunk by chunk using progressively larger subensembles of $\mathscr{T}_k$. For the $l^{\text{th}}$ chunk $\mathcal{D}_{k,l}^S$, the soft label set $Y_{k,l}$ is generated using only the first $l$ teachers in the assigned ensemble: $Y_{k,l} = \mathscr{T}_{k,l}(\mathcal{D}_{k,l}^S)$, where $\mathscr{T}_{k,l} = \cup_{i \in [l]} \mathcal{T}_{k,i}$. This incremental approach further limits the scope of influence of each individual teacher constituent model $\mathcal{T}_{k,i}$ primarily to chunks $l \geq i$.

3. **Slicing:** Each combined data and soft-label pair chunk, denoted $\mathcal{D}_{k,l}^\dagger = [\mathcal{D}_{k,l}^S, Y_{k,l}]$, is then further divided into $R_{k,l}$ disjoint slices $\{\mathcal{D}_{k,l,1}^\dagger, \ldots, \mathcal{D}_{k,l,R_{k,l}}^\dagger\}$ ($\cup_{j \in [R_{k,l}]} \mathcal{D}_{k,l,j}^\dagger = \mathcal{D}_{k,l}^\dagger$, $\cap_{j \in [R_{k,l}]} \mathcal{D}_{k,l,j}^\dagger = \emptyset$), analogous to standard SISA slicing.

This hierarchical structure within PURGE (shards $\rightarrow$ chunks $\rightarrow$ slices) allows for fine-grained checkpointing and efficient unlearning. The student constituent model $\mathcal{S}_k$ is trained incrementally over both chunks and slices, following the SISA principle. Training starts with the first slice of the first chunk ($\mathcal{D}_{k,1,1}^\dagger$) from an initial state $\mathcal{S}_{k,0}$. The model state is checkpointed after completing training for each slice. Let $\mathcal{S}_{k,l,j}$ denote the model state after processing slice $j$ of chunk $l$. To obtain $\mathcal{S}_{k,l,j}$, the preceding state (either $\mathcal{S}_{k,l,j-1}$ if $j > 1$, or $\mathcal{S}_{k,l-1,R_{k,l-1}}$ if $j = 1, l > 1$) is trained for $e_{l,j}$ epochs using the cumulative data processed so far, which includes all data from chunks 1 to $l-1$ plus slices 1 to $j$ of chunk $l$: $(\cup_{i=1}^{l-1} \mathcal{D}_{k,i}^\dagger) \cup (\cup_{q=1}^{j} \mathcal{D}_{k,l,q}^\dagger)$. The loss for a data-label pair $[d, y]$ from a slice is typically a standard distillation loss, $\mathcal{L}(\mathcal{S}_k(d), y)$. This process continues until the final student constituent model $\mathcal{S}_k = \mathcal{S}_{k,c_k,R_{k,c_k}}$ is obtained after processing all chunks and slices derived from shard $\mathcal{D}_k^S$. All intermediate states $\mathcal{S}_{k,l,j}$ are stored to facilitate the fast unlearning process provided by PURGE, as detailed in Section 3.2.

Finally, after all student constituent models $\{\mathcal{S}_1, \ldots, \mathcal{S}_N\}$ are trained via the PURGE framework, a straightforward, non-trainable aggregation function (e.g., averaging the output predictions or logits) is applied during inference to produce the overall output of the student network $\mathcal{S}$. To enhance clarity for the reader, we present the PURGE training procedure in pseudo-code in Appendix A.1.

## 3.2 Unlearning process

The PURGE framework provides mechanisms for efficiently handling unlearning requests targeting either the student's training data or the teacher's training data. A detailed discussion on handling simultaneous unlearning requests for both is provided in the Appendix A.3.

**Unlearning student data** When a request involves removing a student data point $d_u$ (and its corresponding teacher-generated soft label $y_u$) located in slice $\mathcal{D}_{k,l,j}^\dagger$, PURGE follows the standard SISA unlearning procedure for the affected student constituent model $\mathcal{S}_k$. The network state is reverted to the previously saved checkpoint $\mathcal{S}_{k,l,j-1}$ (the state before slice $\mathcal{D}_{k,l,j}^\dagger$ was first processed). Retraining then commences incrementally from this point onwards, using the modified data slice (excluding $[d_u, y_u]$) and all subsequent slices and chunks for that constituent model. This ensures that the unlearning process for student-side data inherits the efficiency benefits of the SISA framework, requiring only partial retraining of a single constituent model.

**Unlearning teacher data** A key challenge addressed by PURGE is handling unlearning requests for data used to train the teacher models. Suppose a data point $d_v$ is removed from the training set originally used for a teacher constituent model $\mathcal{T}_{k,l}$ (which belongs to the ensemble $\mathscr{T}_k$ mapped to student $\mathcal{S}_k$). The unlearning process within PURGE proceeds as follows:

1. The teacher constituent model $\mathcal{T}_{k,l}$ is updated to $\mathcal{T}_{k,l}'$ (presumably using SISA efficiently if the teacher also uses it).

2. All soft labels generated by teacher subensembles that included $\mathcal{T}_{k,l}$ must be updated. This affects chunks $l, l+1, \ldots, c_k$ for student $\mathcal{S}_k$. Specifically, the soft label sets $Y_{k,i}$ (for $i \in [l, c_k]$) need to be regenerated using the updated teacher $\mathcal{T}_{k,l}'$ within the respective subensembles $\mathscr{T}_{k,i}$. This results in updated data-label chunks $\mathcal{D}_{k,i}^{\dagger\prime}$ for $i \in [l, c_k]$.

3. The affected student constituent model $\mathcal{S}_k$ must revert its state. Since the distillation process for chunk $l$ was the first to potentially use $\mathcal{T}_{k,l}$, the student reverts to the state saved just before processing chunk $l$, which is $\mathcal{S}_{k,l-1}$ (equivalent to $\mathcal{S}_{k,l-1,R_{k,l-1}}$).

4. Student $\mathcal{S}_k$ resumes incremental training from chunk $l$ onwards, using the regenerated data-label chunks $\mathcal{D}_{k,i}^{\dagger\prime}$ for $i \in [l, c_k]$ and their constituent slices. The final updated student constituent model is denoted $\mathcal{S}_k'$.

This procedure ensures that the influence of the teacher's unlearned data $d_v$ is removed from the student network $\mathcal{S}_k$, while only requiring partial retraining of that single student constituent model. The corresponding pseudo-code can be found in Appendix A.1.

**Efficiency analysis of teacher unlearning**    We analyze the efficiency gain of PURGE compared to a naive SISA application for teacher-side unlearning. In the naive case, as argued in Section 2, any teacher update requires retraining the entire student network. This takes time equivalent to training the SISA student ensemble from scratch, denoted $t_{\text{sisa}}$. Assuming training time scales linearly with the total number of data points processed (epochs × dataset size), $t_{\text{sisa}}$ is proportional to $e'D$, where $D = |\mathcal{D}^S|$ is the student dataset size and $e'$ is the equivalent number of full-dataset epochs reflecting the total computational effort used for initial training.

For PURGE, only the affected constituent model $\mathcal{S}_k$ retrains partially. We consider an idealized scenario with even distribution: $N$ student constituent models, $M$ teacher constituent models, $c = M/N$ chunks per student shard, and $r$ slices per chunk (total $R = cr$ slices per shard). We assume each slice is trained for $e_R$ epochs, where $e_R = \frac{2}{cr+1}e'$ relates the per-slice epochs to the equivalent full-training epochs $e'$ based on total computational effort [1]. When unlearning affects teacher $\mathcal{T}_{k,l}$, $\mathcal{S}_k$ retrains from chunk $l$ onwards. The average number of slice-processing steps required, $\bar{K}$, averaging over which chunk $l \in [1, c]$ is affected, is derived in the Appendix and given by:

$$\bar{K} = \frac{e_R}{c} \sum_{i=0}^{c-1} \frac{(((ir+1)+cr)((c-i)r))}{2} \tag{1}$$

The retraining time for PURGE, $t_{\text{PURGE}}$, is proportional to the number of slice-processing steps times the number of data points per slice ($\frac{D}{Ncr}$). Thus, $t_{\text{PURGE}} \propto \bar{K}\frac{D}{Ncr}$. The theoretical speed-up of PURGE over naive SISA is then:

$$\frac{t_{\text{sisa}}}{t_{\text{PURGE}}} = \frac{e'D}{\bar{K}\frac{D}{Ncr}} \tag{2}$$

Substituting $e_R = \frac{2}{cr+1}e'$ and the expression for $\bar{K}$ (details in Appendix), this simplifies to:

$$\frac{t_{\text{sisa}}}{t_{\text{PURGE}}} = N \cdot \frac{6c^2r + 6c}{4c^2r + 3cr + 3c - r + 3} \tag{3}$$

As shown in the Appendix (Equation 10), the second factor is greater than 1 for all positive integers $r$ and $c$. Therefore, PURGE provides a speed-up of at least $N\times$ compared to the naive SISA approach for teacher-side unlearning. Expressing this in terms of the total number of teacher constituent models $M = Nc$:

$$\frac{t_{\text{sisa}}}{t_{\text{PURGE}}} = M \cdot \frac{6cr + 6}{4c^2r + 3cr + 3c - r + 3}. \tag{4}$$

This factor decreases as $c$ (chunks per student) increases for fixed $M$ and $r$. This implies that for a fixed number of teachers $M$, the speed-up is maximized when $c$ is minimal (ideally $c = 1$), which corresponds to having more student constituent models ($N = M$).

For unlearning request sent directly to the student side, the efficiency is determined by the total number of slices per shard, $R = c \cdot r$, the product of chunks ($c$) and slices per-chunk ($r$). In this case, the soft-labels do not require updates and the unlearning procedure follows the standard SISA approach. As a result, the expected unlearning cost for a single request can be analyzed as in SISA[1]: it is proportional to $\frac{2}{3} + \frac{1}{3R}$ times the cost of retraining the full shard. Therefore, increasing $r$ leads to a larger $R$ which in turn reduces the unlearning time for student-side requests. The speed-up approaches a maximum of of $1.5\times$ compared to a shard with no slicing. This introduces a trade-off between student-side unlearning speed and the teacher-side unlearning: a larger $r$ will improves student-side unlearning speed but slows down teacher-side unlearning. Consequently, the optimal

Table 1: Datasets and model architectures used in the experiments.

| Dataset | Type | Size | Classes | Model architecture |
|---|---|---|---|---|
| MNIST [8] | Image | 70,000 | 10 | 2 Conv + 2 FC Layers |
| SVHN [12] | Image | 630,420 | 10 | 2 Conv + 2 FC Layers |
| CIFAR-100 [20] | Image | 60,000 | 100 | ResNet50 [14] |
| SST5 [28] | Sentence | 11,855 | 5 | Qwen2.5-7B[31] & BERT[9] |

choice of $r$ depends on the expected ratio of student-side to teacher-side unlearning requests, which is application dependent.

The detailed derivations for Equations 1-4 are presented in the Appendix.

**Rationale for incremental multi-teacher distillation**  A core component of PURGE is the incremental multi-teacher training within each student shard, where the subensemble $\mathscr{T}_{k,l}$ grows as training progresses through chunks $l = 1, \ldots, c_k$. This ensures teacher $\mathcal{T}_{k,i}$'s influence is primarily limited to data processed from chunk $i$ onwards. One might consider an alternative: using only a single, different teacher constituent model $\mathcal{T}_{k,l}$ to generate soft labels $Y_{k,l}^{\text{single}} = \mathcal{T}_{k,l}(\mathcal{D}_{k,l}^S)$ for each chunk $l$. Intuitively, this might seem to isolate influence even further.

However, regarding unlearning efficiency for teacher updates, this alternative offers no advantage. If teacher $\mathcal{T}_{k,l}$ requires unlearning, the student $\mathcal{S}_k$ must still revert to state $\mathcal{S}_{k,l-1}$ and retrain from chunk $l$ onwards, regardless of whether chunk $l$ was trained using only $\mathcal{T}_{k,l}$ or the subensemble $\mathscr{T}_{k,l}$. The number of retraining steps remains the same, and since retraining time is dominated by model training rather than the (usually negligible) difference in inference time for generating soft labels (single vs. subensemble), the overall unlearning time is similar for both approaches.

Crucially, however, the incremental multi-teacher approach provides a more stable training process for the student constituent models. Learning sequentially from potentially diverse single teachers ($\mathcal{T}_{k,1}$, then $\mathcal{T}_{k,2}$, etc.) can introduce abrupt changes in the supervisory signal, destabilizing training. The incremental ensemble $\mathscr{T}_{k,l}$ smooths these transitions by gradually incorporating new teachers while retaining previous ones, leading to better convergence and performance, as demonstrated in our experiments (Section 4). Thus, the incremental multi-teacher strategy is adopted in PURGE.

**Honest Service Provider Assumption**  Our proposed PURGE framework focus on providing verified unlearning guarantees in knowledge distillation settings, addressing unlearning requests that target either the teacher or the student model. When such an unlearning request occurs, the impacted constituent model is reverted to a previously saved checkpoint, before the influence of the removed data, and this data point is fully excluded from the subsequent training. This approach is both auditable and provable: assuming an honest service provider, the authority can audit the code to confirm the proper execution of PURGE training and unlearning, thereby ensuring verified unlearning. While these guarantees may not hold with a dishonest service provider, our proposed method is orthogonal to the methods enforcing the execution of the exact unlearning algorithms.

## 4 Experimental results

We conducted experiments on both image and sentiment classification tasks using the MNIST, SVHN, CIFAR-100 and SST5 (detailed in Table 1) to evaluate the effectiveness of PURGE. The image classification tasks were conducted on a machine equipped with one NVIDIA RTX 3090 GPU (24GB VRAM) while the sentiment classification task was conducted on a machine with 8 NVIDIA A100 GPUs (80GB VRAM each) to support the large number of BERT constituent models.

### 4.1 Unlearning speed evaluation

We first evaluate the speed-up PURGE provides to the student network when unlearning requests target the teacher network's data. Following the setup described in Section 3.2, we assume each slice is trained for the same number of epochs, $e_R$. This is related to the equivalent full-dataset training epochs $e'$ by $e_R = \frac{2}{rc+1}e'$ [1], ensuring comparable total computation during initial training. For this experiment, we set $e' = 120$ and evaluate on the MNIST dataset. We measure the wall-clock

retraining time required for student networks trained using PURGE and compare it against a baseline representing a naive SISA application, where the student network must be fully retrained upon any teacher update. In both cases, the teacher network itself is assumed to use SISA, allowing its own updates to be efficient.

Figure 2 plots the student retraining time against the number of teacher-side unlearning requests processed. For simplicity, we refer to the baseline naive SISA approach as *SISA*. As our focus is the efficiency gain for student retraining provided by PURGE, the times shown exclude the retraining time of the teacher network itself (which is assumed efficient via SISA in both scenarios) and the inference time required by the teacher to generate updated soft labels. While PURGE's constituent mapping allows for faster soft label regeneration compared to the baseline (where the full teacher ensemble always contributes), this inference time is typically negligible compared to retraining time and is thus excluded from our comparison.

We configured the teacher network with $M = 32$ constituent models. For the baseline *SISA* approach, where full student retraining is always required upon a teacher update, the student retraining time is independent of the number of student constituent models $N$. We thus use $N = 8$ constituent models for the baseline measurement. For PURGE, we varied the number of student constituent models $N$ from 1 to 32. We tested two configurations for the number of slices per chunk: $r = 1$ and $r = 4$. Note that while the theoretical analysis uses $e_R = \frac{2}{rc+1}e'$, our practical implementation uses $e_R = \lceil \frac{2}{rc+1}e' \rceil$. This ceiling function means PURGE might perform slightly more computation than the theoretical minimum, particularly for larger $rc$ values (i.e., smaller $N$ or larger $r$), potentially leading to minor deviations from the theoretical speed-up analysis derived using Equations (1-4). We simulated 100 sequential unlearning requests targeting randomly chosen teacher constituent models. For generality, we assume the teacher and student datasets are distinct ($\mathcal{D}_T \not\equiv \mathcal{D}_S$), so only teacher models and subsequently soft labels are updated, while the student dataset $\mathcal{D}^S$ remains unchanged. Figure 2(a) shows the cumulative retraining time, and Figure 2(b) shows the measured average speed-up relative to the baseline *SISA*.

Clearly, for both $r$ values, PURGE's retraining time decreases (and speed-up increases) as the number of student constituent models $N$ increases. With $N = 32$ (where $c = M/N = 1$), PURGE requires averaging $23.17 \pm 0.17$ seconds to handle an unlearning request, achieving $\approx 32\times$ speed-up over the baseline *SISA*, which requires averaging $737.14 \pm 10.08$ seconds for each unlearning request. The empirical average speed-up values in Figure 2(b) closely align with the theoretical prediction from Equation (4) (plotted as the red curve). Minor deviations are consistent with the use of the ceiling function for $e_R$, being slightly more noticeable for $r = 4$ and small $N$ (large $c$), as predicted.

Comparing the results for $r = 1$ and $r = 4$, we observe that a larger $r$ (more slices per chunk) can lead to a slightly smaller speed-up when unlearning teacher data. This stems from the incremental training structure: re-

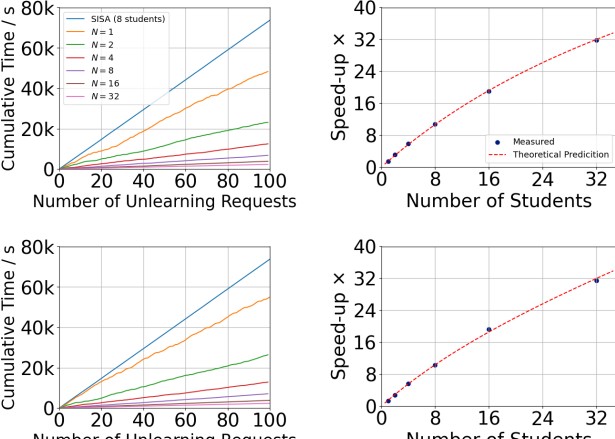

Figure 2: Speed comparison of the student network update process for 100 unlearning requests sent to the teacher network with 32 constituent models ($M = 32$) on MNIST dataset. Top row: $r = 1$; bottom row: $r = 4$ slices per chunk. Left column: cumulative processing time. Right column: measured average speed-up over naive SISA (red curve follows Eq. 4).

training involves recomputing later slices more often (see Eq. 1), and with large $r$, these later slices incorporate more preceding data. Equation (4) confirms this dependency on $r$. It is important to note the trade-off: while larger $r$ might slightly slow down student retraining for teacher unlearning requests, it simultaneously accelerates student retraining for student unlearning requests, because the standard SISA efficiency depends on the total number of slices ($R = cr$). The optimal choice of $r$

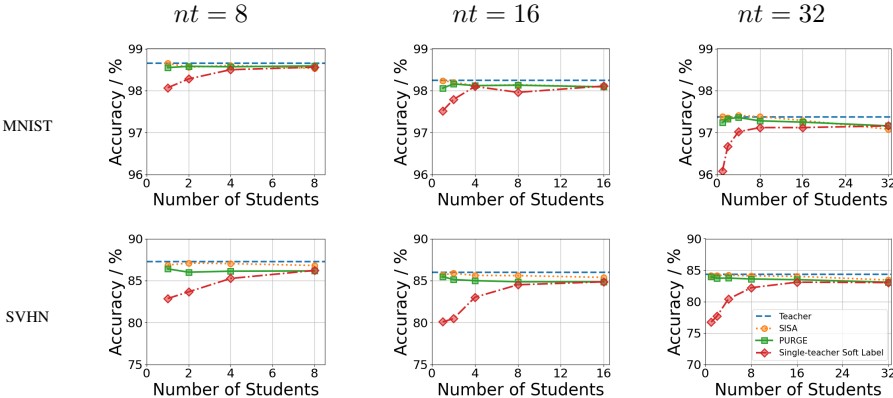

Figure 3: Comparison of student network accuracy on MNIST (top row) and SVHN (bottom row). Accuracy is plotted against the number of student constituent models ($N$) for different teacher ensemble sizes ($M = 8, 16, 32$). The plot shows results for PURGE, the *SISA* baseline student, the original *Teacher* ensemble, and the *Single-teacher Soft Label* ablation.

therefore depends on the expected frequency ratio of teacher versus student unlearning requests for a specific application.

Overall, this experiment demonstrates that PURGE substantially accelerates student network retraining when teacher-side unlearning occurs, validating our theoretical analysis. The efficiency gains scale predictably with the number of student constituent models, confirming the effectiveness of the proposed partitioning and mapping strategy.

## 4.2 Performance evaluation

Having demonstrated PURGE's effectiveness in accelerating retraining, we now evaluate whether these efficiency gains compromise the student network's predictive performance. We compare the performance of student networks trained using our proposed framework against key baselines and an ablation. The baselines are:

- *Teacher*: The original teacher network ensemble, trained using the standard SISA pipeline. This represents an upper bound on performance expected via distillation.

- *SISA*: A student network ensemble trained using the standard SISA pipeline, where each student constituent model learns from its data shard and soft labels generated by the aggregated output of the full *Teacher* ensemble. This represents the naive SISA application to distillation that PURGE aims to outperform in terms of unlearning efficiency.

We also compare PURGE against an ablation using single-teacher soft labels (*Single-teacher*), where soft labels for chunk $l$ are generated only by teacher $\mathcal{T}_{k,l}$. For the multi-teacher aspects within PURGE (and the ablation), we use simple averaging of teacher outputs to generate soft labels. While more advanced multi-teacher distillation techniques exist, averaging serves as a clear baseline for evaluating the structural benefits of PURGE without confounding factors. The datasets and corresponding model architectures (Simple CNN for MNIST/SVHN, ResNet50 for CIFAR-100, BERT and Qwen2.5-7B for SST5) are detailed in Table 1. The experimental results on CIFAR-100 and SST5 are shown in the Appendix.

**Results on MNIST and SVHN** We investigated performance on MNIST and SVHN datasets, varying the number of teacher constituent models $M \in \{8, 16, 32\}$ and, for each $M$, varying the number of student constituent models $N$ from 1 up to $M$ (implying $c = M/N$ chunks per student). Figure 3 presents these results. Both teacher and student networks were trained on the full training sets with data evenly allocated across shards and chunks. The results show that PURGE achieves performance very similar to the baseline *SISA* student, with only a minor degradation compared to the original *Teacher*. For instance, with $M = 32$ teacher constituent models and $N = 32$ student constituent models ($c = 1$), PURGE achieves $97.16\%$ and $83.09\%$ accuracy on MNIST and SVHN respectively, while the *SISA* baseline attains $97.08\%$ and $83.44\%$. This confirms

that PURGE's structural modifications for unlearning efficiency do not significantly compromise predictive performance on these tasks.

As expected, accuracy tends to decline slightly for both PURGE and the *SISA* baseline as the number of student constituent models $N$ increases. This is attributable to the reduced amount of training data available to train each individual constituent model. However, this trend does not hold for the *Single-teacher* ablation (shown in Figure 3). This ablation shows substantial performance degradation compared to the *Teacher* and *SISA* baseline, particularly when $N$ is small (i.e., when each student constituent model learns from many different single teachers sequentially across chunks, $c = M/N$ is large). This performance drop stems from the instability induced by abrupt changes in the supervisory signal when switching between different single teachers for consecutive chunks. In contrast, PURGE's incremental multi-teacher strategy smooths these transitions by gradually incorporating new teachers into the subensemble $\mathscr{T}_{k,l}$, stabilizing the training process. Quantitatively, when learning from an $M = 32$ teacher ensemble with only $N = 1$ student constituent model ($c = 32$), PURGE's accuracy drop relative to the *SISA* baseline is minimal ($0.14\%$ on MNIST, $0.18\%$ on SVHN), whereas the *Single-teacher* ablation suffers significant losses ($1.30\%$ on MNIST, $7.35\%$ on SVHN). This validates the design choice of using the incremental multi-teacher within PURGE for maintaining performance.

## 5    Conclusions

The need for efficient, verifiable machine unlearning is critical, especially for KD used in deploying large models. However, applying verified frameworks like SISA naively to KD is inefficient for teacher-side unlearning, because information propagation forces costly full student retraining, negating SISA's benefits. To address this critical gap, we introduced PURGE (Partitioned Unlearning with Retraining Guarantee for Ensembles), a novel framework integrating the principles of SISA with the specifics of KD. By employing constituent mapping—whereby each teacher constituent model's influence is restricted to specific student constituent models—and utilizing an incremental multi-teacher distillation strategy within each student shard, this framework successfully maintains data isolation throughout the student's training process.

Our theoretical analysis and empirical evaluations on image and language tasks demonstrate that our method achieves its primary objective: it enables efficient, verified unlearning even when teacher data is removed, requiring only partial retraining of the affected student constituent model(s). This results in a substantial speedup (at least $N\times$, where $N$ is the number of student constituent models) compared to the naive baseline. Furthermore, the approach naturally retains SISA's efficiency for handling student-side unlearning requests and, crucially, maintains student performance comparable to the standard SISA baseline, validating the stability provided by the incremental multi-teacher strategy.

By ensuring efficient and verified unlearning within teacher-student pipelines, this capability makes the responsible deployment and maintenance of distilled models significantly more practical, particularly for systems involving large foundation models as teachers. Future research could explore several promising directions: integrating more sophisticated multi-teacher distillation algorithms within this structure to potentially enhance student learning efficiency and final performance; extending the theoretical analysis to cover different data distributions or aggregation methods; and applying and evaluating the framework in complex, large-scale distillation scenarios involving state-of-the-art vision or language models.

## Acknowledgments

This research was financially supported by Engineering and Physical Sciences Research Council (EPSRC), United Kingdom, [SUSTAIN Manufacturing Hub EP/S018107/1].

GM acknowledges support from a UKRI AI Turing Acceleration Fellowship (EPSRC EP/V024868/1).

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

# A   Supplemental material

## A.1   Pseudo-code For PURGE

**Algorithm 1** Algorithmic outline of the PURGE training framework

---
1: **for** $k = 1$ to $N$ **do**
2:     Divide $\mathcal{D}_k^S$ into $c_k$ ordered, disjoint chunks $\{\mathcal{D}_{k,1}^S, \ldots, \mathcal{D}_{k,c_k}^S\}$
3:     Initialize student model state $\mathcal{S}_{k,0}$
4:     **for** $l = 1$ to $c_k$ **do**
5:         Generate soft labels $Y_{k,l}$ for chunk $\mathcal{D}_{k,l}^S$ using teachers $\mathscr{T}_{k,l} = \cup_{i=1}^l \mathcal{T}_{k,i}$
6:         Pair chunk data and labels: $\mathcal{D}_{k,l}^\dagger = [\mathcal{D}_{k,l}^S, Y_{k,l}]$
7:         Partition $\mathcal{D}_{k,l}^\dagger$ into $R_{k,l}$ slices $\{\mathcal{D}_{k,l,1}^\dagger, \ldots, \mathcal{D}_{k,l,R_{k,l}}^\dagger\}$
8:         **for** $j = 1$ to $R_{k,l}$ **do**
9:             Select previous state: $\mathcal{S}_{k,l,j-1}$ if $j > 1$; else $\mathcal{S}_{k,l-1,R_{k,l-1}}$ if $l > 1$; else $\mathcal{S}_{k,0}$
10:             Gather cumulative data: $(\cup_{i=1}^{l-1}\mathcal{D}_{k,i}^\dagger) \cup (\cup_{q=1}^j \mathcal{D}_{k,l,q}^\dagger)$
11:             **For** $e_{l,j}$ epochs, **do**:
12:                 Compute distillation loss $\mathcal{L}(\mathcal{S}_k(d), y)$ for each $[d, y]$ in cumulative data
13:                 **Update model parameters** via gradient descent
14:             Store updated intermediate state $\mathcal{S}_{k,l,j}$ for efficient unlearning
15:         **end for**
16:     **end for**
17:     Final model for shard: $\mathcal{S}_k = \mathcal{S}_{k,c_k,R_{k,c_k}}$
18: **end for**

---

**Algorithm 2** Algorithmic outline for student-side unlearning when unlearning request sent to the teacher-side in PURGE

---
1: Update teacher constituent model: $\mathcal{T}_{k,l}$ is the current teacher constituent model
2: $\mathcal{T}_{k,l}'$ is obtained by SISAUpdate($\mathcal{T}_{k,l}$, remove data point $d_v$)
3: **for** $i = l$ to $c_k$ **do**
4:     Generate soft labels $Y_{k,i}$ for chunk $i$ using updated teachers $\mathscr{T}_{k,i}$ (including $\mathcal{T}_{k,l}'$)
5:     Pair chunk data and labels: $\mathcal{D}_{k,i}^{\dagger'} = [D_{k,i}^S, Y_{k,i}]$
6: **end for**
7: Revert student constituent model: set $\mathcal{S}_{k,l-1}$ to checkpoint before chunk $l$
8: Initialize $\mathcal{S}_k' \leftarrow \mathcal{S}_{k,l-1}$
9: **for** $i = l$ to $c_k$ **do**
10:     Incrementally train $\mathcal{S}_k'$ using $\mathcal{D}_{k,i}^{\dagger'}$ and slices
11:     Update and store intermediate state $\mathcal{S}_{k,i}'$
12: **end for**
13: **return** $\mathcal{S}_k'$

---

## A.2   Proof of speed-up when teacher unlearns

We follow the same setup for the epoch number as done in SISA with the same number of epochs $e_R$ for each round of training on all slices. With incremental training applied on the slice level, with training progressing through the slices, the number of data points to cover in one epoch is increasing. We consider evenly distributed data, slices, chunks, and shards for a student network with student constituent models $\{\mathcal{S}_i\}$. When the $l^{\text{th}}$ teacher for the $k^{\text{th}}$ student is changed, the student constituent model $\mathcal{S}_k$ will reverse back to $S_{k,l-1}$ and retrained from $(l-1)r + 1^{\text{th}}$ chunk to $cr^{\text{th}}$ chunk. Thus, the total number of slices the retraining process needs to run is:

$$K(l) = \sum_{j=(l-1)r+1}^{cr} \sum_{i=1}^j e_R = e_R \frac{(((l-1)r+1) + cr)(cr - (l-1)r)}{2}, \tag{5}$$

where $c$ is the number of chunks per shard and $r$ is the number of slices per chunk.

Assuming that every teacher shares the same probability of receiving an unlearning request, the average number of data points for a student to retrain when an unlearning request is sent to one teacher can be calculated as:

$$
\begin{aligned}
\bar{K} &= \frac{1}{c} \sum_{l=1}^{c} K(l) \\
&= \frac{1}{c} \sum_{l=1}^{c} e_R \frac{(((l-1)r+1)+cr)(cr-(l-1)r)}{2} \\
&= \frac{e_R}{c} \sum_{l=0}^{c-1} \frac{((lr+1)+cr)((c-l)r)}{2}
\end{aligned}
\tag{6}
$$

With the assumption that the training time is solely dependent on the amount of training data and each network is initially trained for equal time from scratch, we consider a data set of size $D$ trained for $e'$ epochs. As the proposed framework should be trained for equal time, for a network with $N$ student constituent models, this gives:

$$
e'D = ND_{\text{slice}} \sum_{i=1}^{cr} e_R = N \frac{D}{Ncr} \sum_{i=1}^{cr} e_R = e_R \frac{(cr+1)D}{2},
\tag{7}
$$

where $D_{\text{slice}} \frac{D}{Nrc}$ is the number of data points in one slice and $\sum_{i=1}^{cr} e_R$ is the total number of slices a student constituent model would run through over the entire training process. It can be easily derived that

$$
e_R = \frac{2}{cr+1} e'.
\tag{8}
$$

Given that SISA requires full retraining when an unlearning request is sent to a teacher constituent model and requires training on $e'D$ data points, the ratio of retraining time for SISA against the proposed method can be computed as:

$$
\begin{aligned}
\frac{t_{\text{SISA}}}{t_{\text{couple}}} &= \frac{e'D}{\bar{K}D_{\text{slice}}} \\
&= \frac{e'D}{\frac{e_R}{c} \sum_{i=0}^{c-1} \frac{((lr+1)+cr)(c-l)r}{2} \frac{D}{Ncr}} \\
&= \frac{1}{\frac{1}{c} \frac{2}{cr+1} \sum_{i=0}^{c-1} \frac{((lr+1)+cr)(c-l)r}{2} \frac{1}{Ncr}} \\
&= N \cdot \frac{6c^2r + 6c}{4c^2r + 3cr + 3c - r + 3}
\end{aligned}
\tag{9}
$$

It can be shown that the proposed method provides at least $N\times$ speed-up by showing that the second part of the expression in Equation (9) is bigger than 1:

$$
\begin{aligned}
\frac{6c^2r + 6c}{4c^2r + 3cr + 3c - r + 3} - 1 &= \frac{2c^2r + 3c - 3cr + r - 3}{4c^2r + 3cr + 3c - r + 3} \\
&= \frac{r(2c-1)(c-1) + 3(c-1)}{r(4c-1)(c+1) + 3(c+1)} \\
&\geq 0 \qquad \qquad \because c \geq 1
\end{aligned}
\tag{10}
$$

For evenly distributed chunks, we have $N = M/c$. From Equation (10), we can write the ratio for speed-up in terms of the number of teacher constituent models $M$ and the number of chunks per student $c$ as:

$$
\frac{t_{\text{SISA}}}{t_{\text{couple}}} = M \cdot \frac{6cr + 6}{4c^2r + 3cr + 3c - r + 3}.
\tag{11}
$$

For the second part, its derivative with respect to $c$ is:

$$
\frac{d(\frac{6cr+6}{4c^2r+3cr+3c-r+3})}{dc} = -\frac{6(r^2(4c^2+1) + 8cr + 3)}{(c+1)^2(r(4c-1)+3)^2},
\tag{12}
$$

which is negative for all positive integer $r$ and $c$. Thus, we will have less speed-up when we have more chunks for each student constituent model. When the number of teacher constituent models is fixed, this means that by having more student constituent models, we can have a faster retraining process.

### A.3 Handling simultaneous unlearning requests

In Section 3, we discussed how the proposed PURGE can address individual unlearning requests directed at the student's or the teacher's data. In scenarios where the teacher and student potentially share the same underlying dataset ($\mathcal{D}^T \equiv \mathcal{D}^S$, common in self-distillation or when using a public dataset), an unlearning request might require removing a data point $d_u$ simultaneously from both networks. The PURGE framework efficiently handles this as well. We consider two main cases:

**Scenario 1: aligned data removal.** Suppose the data point $d_u$ to be removed exists in student slice $\mathcal{D}^{\dagger}_{k,l,j}$ and also corresponds exactly to data originally used to train the teacher constituent model $\mathcal{T}_{k,l}$ (i.e., the teacher responsible for generating labels starting from chunk $l$ in the affected student constituent model $\mathcal{S}_k$). The unlearning process can be combined:

1. Teacher $\mathcal{T}_{k,l}$ updates to $\mathcal{T}'_{k,l}$.

2. Soft labels $Y_{k,i}$ for $i \in [l, c_k]$ are regenerated using updated subensembles including $\mathcal{T}'_{k,l}$.

3. The student constituent model $\mathcal{S}_k$ reverts to state $\mathcal{S}_{k,l-1}$ (due to the teacher change affecting chunk $l$ onwards).

4. $\mathcal{S}_k$ retrains incrementally from chunk $l$ onwards, using the updated soft labels *and* excluding $d_u$ from the relevant student slice $\mathcal{D}_{k,l,j}$. The effective slice becomes $\mathcal{D}'_{k,l,j} = \mathcal{D}_{k,l,j} \setminus \{[d_u, y_u]\}$.

The retraining time is dominated by the steps required for the teacher update (reverting to $\mathcal{S}_{k,l-1}$) and is nearly identical to handling only the teacher unlearning request, ignoring the negligible effect of removing one data point from the student's retraining path.

**Scenario 2: misaligned data removal** Suppose $d_u$ is removed from student slice $\mathcal{D}^{\dagger}_{k,l,j}$, but the corresponding data point's removal affects a *different* teacher constituent model $\mathcal{T}_{\mu,\nu}$, where $(k,l) \neq (\mu,\nu)$. In this case, two separate unlearning processes occur concurrently:

1. Student constituent model $\mathcal{S}_k$ handles the removal of $d_u$ using the standard SISA process: revert to $\mathcal{S}_{k,l,j-1}$ and retrain onwards.

2. Student constituent model $\mathcal{S}_\mu$ handles the update resulting from teacher $\mathcal{T}_{\mu,\nu}$ unlearning its data, following the PURGE process for teacher unlearning: revert to $\mathcal{S}_{\mu,\nu-1}$ and retrain onwards using updated soft labels.

The total unlearning time is approximately the sum of the times for these two independent partial retraining processes. In both scenarios, PURGE requires only partial retraining of at most two student constituent models. This contrasts sharply with a naive SISA application, where any teacher update (as occurs in both scenarios) would necessitate full retraining of all $N$ student constituent models. Therefore, PURGE offers substantial efficiency gains even when handling simultaneous unlearning requests.

### A.4 Training Stability: Multi-Teacher vs. Single-Teacher Soft Labels Approaches

Our experimental results demonstrate that the incremental multi-teacher learning strategy introduced by the PURGE framework consistently outperforms the single-teacher soft label ablation approach. This advantage primarily stems from the enhanced training stability provided by multi-teacher soft label generation. Unlike conventional epoch-based methods, SISA-based unlearning frameworks employ incremental training over data slices, enabling checkpointing at the point each data slice is first introduced. While this design allows efficient backtracking to specific checkpoints for unlearning requests, it introduces potential instability. Specifically, early training rounds on smaller data slices may cause the model to overfit to those initial slices. Subsequently, when a new slice is added, the

model's loss can fluctuate more than in standard epoch-based training, since it must train on both previously learned and new data, resulting in greater gradient variation and increased instability.

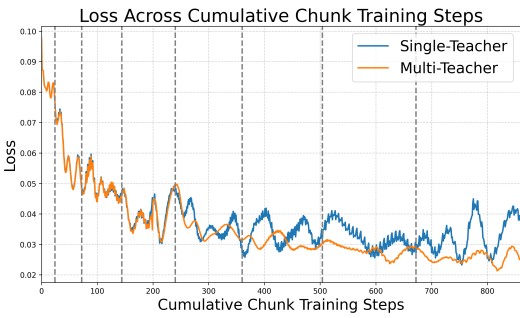

Figure 4: Loss curve comparison between multi-teacher soft label generation and single-teacher ablation

The PURGE framework introduces additional data chunk partitioning to enable precise mapping between teacher and student constituent models. This design exacerbates instability in the single-teacher soft-label setup: the model must simultaneously adapt to both newly introduced data and an unfamiliar teacher constituent model responsible for generating the corresponding soft labels. To illustrate this effect, Figure 4 plots the loss curve during the training of a student constituent model, showing the dynamics when learning from a shard comprised of data chunks with soft labels produced by 8 teacher constituent models.

The blue curve in Figure 4 corresponds to the single-teacher ablation while the orange curve being the proposed multi-teacher soft label generation. We plot the loss against the cumulative chunk training steps, counting every instance a chunk is trained (including repeats). The vertical dashed lines are the indicators when a new data chunk is introduced to the incremental training process. Except for the first interval, which involves training on a single chunk, each subsequent chunk training step selects a different chunk for training. This process cycles through all available chunks in turn, ensuring that each chunk is trained for exactly $e_R$ epochs before the next chunk is introduced.

From the plot, we observe that during the first three intervals, the loss curves for the single-teacher ablation and the multi-teacher soft-label training are quite similar, as the training process cycles through only a small number of data chunks. However, as training progresses and more chunks are introduced, the single-teacher ablation exhibits significantly larger fluctuations. This increased instability arises because the model alternates between data chunks that have already been trained extensively and newly added chunks. The newly introduced chunks not only provide previously unseen inputs, but their corresponding soft labels generated by a newly introduced teacher model are also unfamiliar to the student. This unfamiliarity can result in pronounced changes in the loss and, consequently, larger gradients, leading to an unstable training process.

In contrast, our multi-teacher soft-label generation approach mitigates such fluctuations. Although the data may still be new to the student model, the soft labels are produced by averaging predictions across multiple teacher models from prior training steps. This aggregation moderates sharp changes and smooths the training dynamics. As a result, while some fluctuations are still present, their magnitude is much lower than in the single-teacher ablation, providing a more stable training process and ultimately better model performance.

## A.5 Performance on distillation with smaller student training dataset

In knowledge distillation, it is common for the student to be trained on a smaller subset of the data distilled by the teacher. With less overall training data, each student constituent model will receive a smaller data allocation. Consequently, the balance between increasing the number of constituent models for faster unlearning and ensuring each constituent model is well-trained becomes more critical. To investigate how the proposed method may perform under such conditions, we conduct experiments on the $10\%, 20\%$, and $50\%$ subsets of MNIST and SVHN. The experimental results show that PURGE produces similar performance to *SISA* under all conditions, with more stable performance compared to *Single-teacher Soft Label* when each student constituent model learning from a large number of teacher constituent models. Overall, the effectiveness of PURGE is demonstrated.

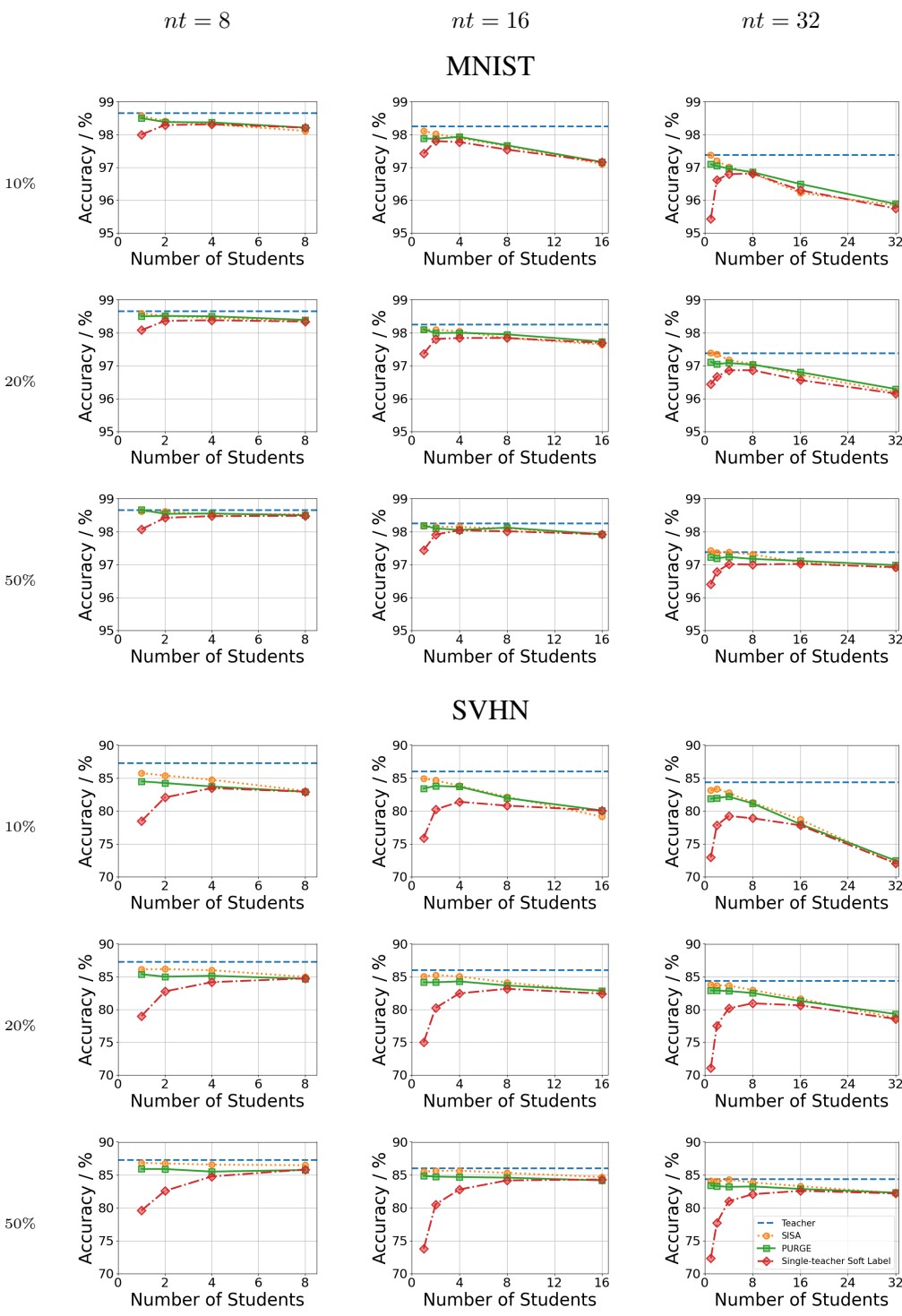

Figure 5: Comparison of student network accuracy on 10%, 20% and 50% versions of MNIST and SVHN. The plot shows results for PURGE, the *SISA* baseline student, the original *Teacher* ensemble, and the *Single-teacher Soft Label* ablation. The original *Teacher* ensemble was trained on the full training sets, while the student networks are trained on the corresponding subsets.

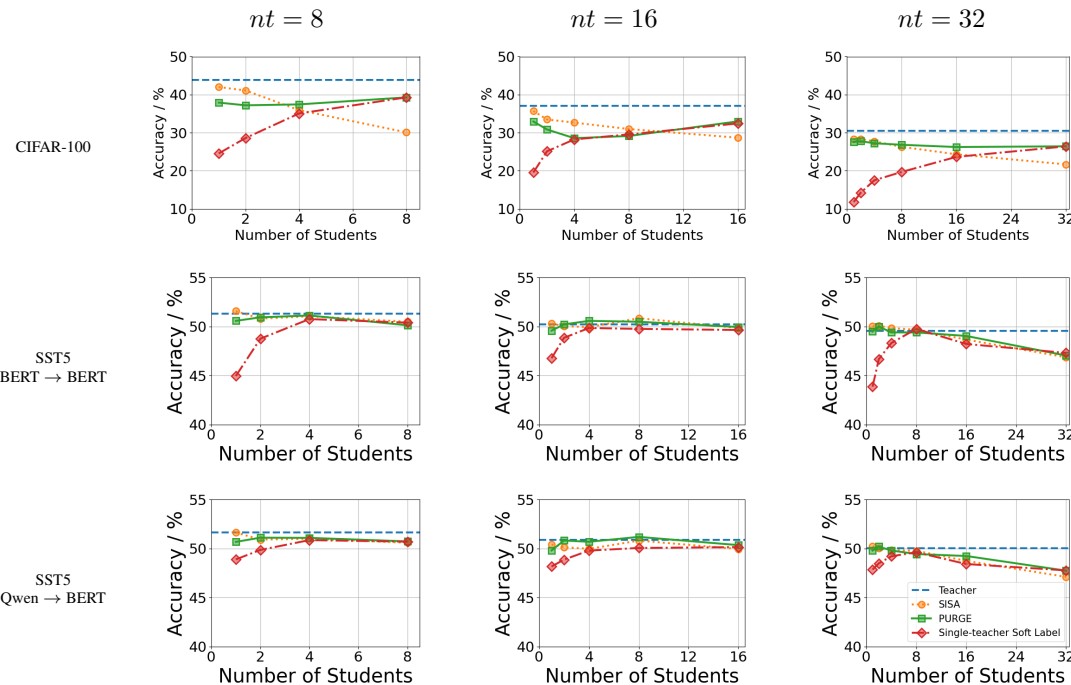

Figure 6: Comparison of student network accuracy on CIFAR-100 (top row) and SST5 (mid and bottom rows). For the SST5 experiment, we employed both BERT and Qwen2.5-7B as teacher models, distilling their knowledge into BERT student models. Accuracy is plotted against the number of student constituent models ($N$) for different teacher ensemble sizes ($M = 8, 16, 32$). The plot shows results for PURGE, the *SISA* baseline student, the original *Teacher* ensemble, and the *Single-teacher Soft Label* ablation. Models were trained on the full training sets of CIFAR0-100 and SST5.

## A.6 Experimental Results: CIFAR-100, SST-5, and Large Language Models (Qwen2.5 7B/32B)

We evaluated performance on the more challenging CIFAR-100 dataset for image classification using ResNet50 and the SST5 dataset for sentiment analysis using BERT. Figure 6 shows the performance of PURGE compared to the baselines under these conditions.

The complexity of the image classification task on CIFAR-100 is reflected in the relatively low accuracy of the *Teacher* ensemble itself, which achieved $30.5\%$ on the test set with a teacher ensemble of $M = 32$ constituent models. Despite the task's difficulty and the performance gap relative to the original *Teacher*, PURGE demonstrates effectiveness by achieving performance very comparable to the *SISA* baseline student across different numbers of student constituent models ($N$). As seen in Figure 6, the accuracy of PURGE closely follows that of the *SISA* baseline while significantly outperforms the *Single-teacher Soft Labels*, particularly when the number of students is small. When the number of students is large, especially when the teacher and student have the same number of constituent models, $M = N$, we see the proposed PURGE outperforms baseline SISA.

This highlights that for complex tasks, the constituent mapping introduced in the proposed PURGE framework enables each student constituent model to learn fine-grained information contained in the soft labels produced by its corresponding teacher sub-ensemble of constituent models. In contrast, the baseline SISA aggregates teacher outputs, which can obscure these detailed signals and lead to reduced performance. The results also highlight the broader challenges of ensemble learning in multi-teacher distillation, with such challenges also existing for the proposed PURGE framework since soft-label aggregation still occurs within each teacher sub-ensemble. A promising direction for future work is to explore more refined multi-teacher aggregation strategies while preserving the data isolation between teacher constituent models as established in PURGE.

Although sentiment analysis is a different task from image classification, the results on the SST5 dataset shown in Figure 6 exhibit a similar trend to those observed in image-based tasks. In this experiment, we trained two versions of the teacher model: one based on BERT and the other on Qwen2.5-7B. Both teacher models were trained for $e' = 20$ epochs, and in both cases, distilled to BERT student models. Interestingly, the results of the BERT and Qwen-based experiments are highly comparable, with the more capable Qwen2.5-7B teacher delivering only modest improvements over the BERT teacher. This minor performance difference suggests that the limiting factor may be the sharding strategy used in the SISA framework, rather than architectural differences or model scale. Notably, the largest difference between the two scenarios arises in the results for the single-teacher ablation, where Qwen2.5-7B–distilled students outperform their BERT-based counterparts, demonstrating that single-teacher setups may experience larger performance drops, even when the teacher model itself differs slightly. By contrast, the proposed PURGE framework maintains stable performance across both version of the experiment. In both cases, PURGE demonstrates performance comparable to SISA, while the incremental multi-teacher distillation approach proves more effective than using single-teacher soft labels alone. Combined with the results on CIFAR-100, these findings confirm PURGE's viability for moderately complex tasks, its scalablility to larger model like Qwen2.5-7B, and its general applicability across different domains, preserving the accuracy of standard SISA distillation while introducing substantial unlearning efficiency.

To further extend our evaluation beyond sentiment analysis, we applied the PURGE framework to the more demanding domain of language reasoning on the ARC dataset[7], employing Qwen2.5-32B as the teacher model and distilling into Qwen2.5-3B student models. Using an 8-teacher, 8-student configuration and LoRA [16] fine-tuning, we compared PURGE with naive SISA on ARC-challenge four-choice questions. In this scenario, the teacher only achieved $27.76\%$ accuracy, and both PURGE and naive-SISA exhibited limited student performance, $26.09\%$ and $26.42\%$ accuracy, respectively. It reflects the further constraints imposed by sharding strategies, which limits the data availability for each constituent model, in complex reasoning tasks. As a result, even the teacher merely surpasses chance levels, and student models trained under both frameworks struggle to provide high accuracy. These findings underscore a pressing need for future work to achieve stronger results while maintaining strict data isolation required for sharding-based verified unlearning.

Despite these accuracy limitations, our analysis of unlearning efficiency highlights the practical value of PURGE. Leveraging an 8 Nvidia A100 GPU setup, the PURGE framework enabled student-side updates in just 2.36 GPU-hours for teacher-side unlearning requests, compared to 18.9 GPU-hours for naive-SISA. Given that each teacher checkpoint occupies 128GB and each student checkpoint uses 12GB, PURGE does require storage of multiple checkpoints for each student constituent model. For example, in our experimental setup with eight student constituent models, each composed of one chunk and four slices per chunk, the overall storage requirement for student checkpoints reached 386GB. In similar distillation settings, this additional storage demand for student models remains well within the capabilities of typical server infrastructures. The trade-off between increased storage requirements and accelerated unlearning can be flexibly managed according to the specific demands of the application, taking into account available storage resources and the expected frequency of unlearning requests. More importantly, PURGE achieves a substantial reduction in computational time and cost for verified unlearning in large-scale deployments, outperforming both naive-SISA and conventional retraining from scratch. Thus, even as scaling challenges remain for complex tasks, the PURGE framework demonstrates significant improvements in computational resource efficiency for unlearning, positioning it as a practical approach for efficient, privacy-preserving model deployment.

### A.7 Data Mapping

Data mapping is a critical component of sharding-based unlearning frameworks. In this work, we use uniform random partitioning as it provides generality, prevents systematic bias, and scales well by enabling uniform hyperparameters across constituent models.

While uniform partitioning is the default, alternative strategies are possible. Notably, distribution-aware sharding, as demonstrated in the original SISA framework, groups data points that are more likely to be unlearned into dedicated shards. This can optimize unlearning speed in scenarios where such prior knowledge is available and our PURGE framework is fully compatible with such alternatives.

In addition to distribution-aware sharding, our study of the student–teacher knowledge distillation framework also raises important questions regarding how data is mapped between students and teachers, particularly when faced with imbalanced data distributions. To investigate this, we conducted experiments on the SST-5 dataset focusing on label-imbalanced partitioning. In this setting, each teacher shard was deliberately structured to have one class overrepresented and the others underrepresented, with a fixed class ratio of 0.24:0.19:0.19:0.19:0.19 across the five sentiment categories. For each of the five teacher shards, one class was assigned the largest proportion (0.24) while the rest were each allocated 0.19, and the final shard contained any remaining samples cover the whole training dataset.

Table 2: Impact of label-imbalanced partitioning on teacher and student performance under matched and mismatched data mappings on SST-5. The reported values are the model accuracy.

| No. of students | 1 | 2 | 4 | 8 |
|---|---|---|---|---|
| Matched (Single) | 45.02% | 48.57% | 49.59% | 49.77% |
| Matched (PURGE) | 50.59% | 51.02% | 51.87% | 49.77% |
| Mismatched (Single) | 45.10% | 48.78% | 49.59% | 25.34% |
| Mismatched (PURGE) | 50.72% | 50.90% | 51.95% | 25.34% |
| Teacher Baseline | | 48.86% | | |

We explored two main scenarios: a **"matched"** setup, where each student was trained on a data shard exhibiting the same class imbalance as its corresponding teacher, and a **"mismatched"** configuration, where students received shards with different imbalanced class distributions. In both scenarios, we trained the models using our proposed PURGE framework with the multi-teacher soft label strategy and, for comparison, the single-teacher soft label ablation. Each approach utilized eight teacher constituent models and varied the number of student models. Both the teacher and student models are implemented using BERT. Our results (Table 2) demonstrate that under matched conditions, the accuracy of teacher models fell from $51.32\%$ (with uniform partitioning as shown in Figure 6) to $48.86\%$ with the imbalanced class ratio. This decline was even more pronounced following 1-to-1 distillation, with student accuracy sharply dropping to $25.34\%$ by training 8 student constituent models, each paired with a single teacher. This significant degradation highlights how even relatively mild label imbalances in partitioning can be strongly amplified by the distillation process, particularly in small datasets like SST-5, where each shard contains approximately 1,000 examples.

However, the impact of label imbalance is substantially mitigated once we move away from the strict matching between teacher and student distributions. Training each student on a mismatched data chunk led to a significant recovery in performance, with student accuracy rising to $48.77\%$. Importantly, once students learn from multiple teachers, the performance gap between matched and mismatched scenarios becomes minimal for both single- and multi-teacher soft label generation setups. Furthermore, aggregating soft labels from multiple teachers further enhanced resilience to bias. For instance, in the eight-teacher, four-student configuration, mismatched student accuracy reached $51.95\%$, closely matching the $51.84\%$ achieved under matched conditions, and both outperformed the single-teacher ablation baselines ($49.68\%$ and $49.59\%$ for mismatched and matched, respectively). These trends validate the robustness of incremental multi-teacher distillation, which consistently outperforms single-teacher setups and maintains data isolation between teacher shards.

The results show that while label imbalance can significantly amplify bias and degrade performance under strict matching, this effect is mitigated by mismatched mappings or by aggregating information from multiple teacher constituent models. Consequently, the choice of partitioning strategy could be guided by the application context and available prior knowledge, with uniform random partitioning serving as a robust default and alternative strategies offering potential efficiency gains and maintain performance when appropriately applied.

## A.8 Broader Impacts

Artificial intelligence has become deeply integrated into many aspects of society, with generative models, large language models (LLMs), and other AI tools significantly boosting societal productivity. However, training these models, particularly LLMs, can incur substantial costs. For instance, Sam Altman stated that the cost of training *GPT-4* was more than $100 million.

Beyond financial cost, the environmental impact is also significant. Luccioni *et al.*[23] estimated the training process of BLOOM, a 176B parameter language model, contributed to around 24.7 tonnes of carbon dioxide equivalent ($CO_2$eq) emissions. Therefore, reducing unnecessary model training brings both economic and environmental benefits.

Verified unlearning methods are designed to safeguard user data privacy by ensuring that once a data removal is requested, all influence of that data is effectively and provably eliminated from the trained model. This strict compliance with data regulations like GDPR and CCPA, which grant individuals the right to request their personal data to be erased, requires more than simply deleting data from the training set. It requires the guarantee that the model no longer retains any information derived from the removed data. By enabling efficient model updates without the need for full retraining, verified unlearning safeguards user data privacy, meets the rigorous legal standards, but also provides clear economic and environmental benefits in the process, particularly in real-world scenarios where a large volume of unlearning requests may occur.

Our proposed PURGE framework is specifically designed for verified unlearning in the context of knowledge distillation (KD), which is widely used in domain adaptation and model compression for local deployment of machine learning models, e.g., the distillation of models like ChatGPT. Existing verified unlearning methods typically require full retraining of the student model whenever the teacher model is updated, due to the lack of solutions tailored to the student-teacher paradigm. PURGE addresses this gap by enabling efficient model updates in response to unlearning requests on either the teacher or student side. As KD is common in practice, the proposed PURGE framework can significantly reduce the economic cost and alleviate the environmental impact associated with the retraining process. Thereby, the proposed PURGE can contribute to more sustainable and privacy-compliant AI deployment, bringing broader societal impact through responsible and efficient use of machine learning technology.

