# OpenReview forum: "Efficient Verified Unlearning For Distillation"
_NeurIPS.cc/2025/Conference — NeurIPS 2025 poster_

### Official Review · Reviewer_zLhe · 2025-06-16

**Clarity:** 4
**Significance:** 2
**Originality:** 2
**Rating:** 2
**Confidence:** 4

**Summary:**

This paper proposes PURGE (Partitioned Unlearning with Retraining Guarantee for Ensembles), an unlearning framework tailored for knowledge distillation. The method proposes a student-teacher constituent mapping and an incremental multi-teacher distillation strategy to ensure data isolation during the distillation process. The approach extends SISA to allow partial student retraining when teacher-side unlearning occurs, avoiding the inefficiencies of full student retraining. Experiments demonstrate that PURGE achieves significant speed-ups in unlearning while maintaining performance.

**Questions:**

1. Machine unlearning refers to removing the influence of specific training data from a trained model. In this work, the student-side unlearning operations are more accurately characterized as "cascading synchronization/update after unlearning" rather than unlearning. This distinction is conceptually important. Could the authors clarify how they define unlearning and what distinguishes the proposed approach from standard SISA beyond the cascading synchronization mechanism?

2. The term "verified unlearning" is used repeatedly throughout the paper. Could the authors provide a precise definition of what constitutes "verified unlearning" in this context, along with the formal criteria or standardized references that support this terminology? Additionally, what theoretical proofs or empirical guarantees demonstrate that the resulting model is equivalent to one trained without the deleted data, thereby justifying the "verified" claim?


3. The paper assumes SISA as the default verified unlearning framework without questioning its scalability limitations, such as high training cost from maintaining multiple models and performance degradation due to data fragmentation. Is the sharding architecture truly necessary for machine unlearning, or could alternative approaches achieve similar goals without these inherent overhead costs?

4. Have the authors considered the practical utility of approximate unlearning methods, and could they provide a comparative analysis of the trade-offs between exact and approximate approaches in terms of performance, efficiency, and regulatory compliance?

**Ethical Concerns:**

["NO or VERY MINOR ethics concerns only"]

**Final Justification:**

Due to insufficient contributions, I will maintain my score.

1) The use of SISA to teacher and student models is overly simplistic. The contribution is limited to a mapping between teacher and student models.

2) The understanding Verified Unlearning are fundamental misunderstanding. No verification process/effort was found in the claimed "Verified Unlearning".

3) The work inherits all limitations of SISA.

**Limitations:**

Yes

**Paper Formatting Concerns:**

No major formatting issues observed.

**Quality:**

2

**Strengths And Weaknesses:**

Quality：
The method is sound and builds on SISA, but it inherits SISA’s scalability issues, maintaining many models and checkpoints is costly. Experiments are limited to small datasets, with no substantial evaluation on large models like BERT.


Clarity: The writing is clear and well-organized. Key ideas like constituent mapping and incremental distillation are explained logically and are easy to follow.

Significance:
The paper defines the student-side retraining as part of machine unlearning, but this is conceptually questionable. Machine unlearning aims to remove training data influence from a trained model, whereas the student model here never directly sees the data being unlearned. The proposed method is better viewed as a pipeline update mechanism, NOT unlearning.

Originality:
Although the paper explores a new angle in distillation, it does not introduce a fundamentally new unlearning method, as it is entirely based on SISA. Since the student model does not directly unlearn any data, the work falls outside the core definition of machine unlearning.

---

> ### Author Rebuttal · Authors · 2025-07-29
>
> We thank the reviewer for their detailed and thought-provoking feedback. You have raised several important conceptual questions that go to the heart of how unlearning is defined in complex systems like knowledge distillation. We are grateful for the opportunity to clarify our work's formal guarantees and demonstrate its significance as a necessary advancement in the verified unlearning landscape.
>
> **Apologies for any inconsistency between mathematical notation and LaTeX expressions here. Some symbols may not render correctly.**
>
> **On the definition of unlearning in a distillation pipeline.** We thank the reviewer for this nuanced question. We would like to clarify why we firmly believe this work falls within the scope of machine unlearning. The core issue, as we state in our introduction, is that "information about the teacher's training data can leak and propagate pervasively throughout the student network"  during distillation. Even if a data point $d$ is perfectly unlearned from a teacher $\mathcal{T}$, its influence may persist in any student $\mathcal{S}$ already trained by $\mathcal{T}$. Our work addresses the critical and non-trivial task of verifiably removing this leaked influence from the student.
>
> This process is more rigorous than a "cascading synchronisation." A simple update might involve just swapping the teacher model. Our method, in contrast, adheres to the core principle of SISA-based unlearning: when a teacher constituent $\mathcal{T}_{k,l}$ is updated, the student $\mathcal{S}_k$ is reverted to a checkpoint `$\mathcal{S}_{k,l-1}$` that was saved *before* any influence from `$\mathcal{T}_{k,l}$` was introduced. This rewind-and-retrain procedure is a fundamental unlearning operation, not merely a pipeline synchronisation.
>
> To formalise this, let $\mathcal{A}$ be the training algorithm. A teacher $\mathcal{T}$ is trained on its data $\mathcal{D}^T$, so $\mathcal{T} = \mathcal{A}(\mathcal{D}^T)$. A student $\mathcal{S}$ is then trained via distillation, making it a function of the teacher's outputs and its own data $\mathcal{D}^S$: $\mathcal{S} = \mathcal{A}(\mathcal{T}(\mathcal{D}^S), \mathcal{D}^S)$. When a point $d$ is removed from $\mathcal{D}^T$, the teacher is updated to $\mathcal{T}' = \mathcal{A}(\mathcal{D}^T \setminus d)$. The goal of student-side unlearning is to produce a new student $\mathcal{S}'$ such that $\mathcal{S}' = \mathcal{A}(\mathcal{T}'(\mathcal{D}^S), \mathcal{D}^S)$.
>
> This ensures the final student $\mathcal{S}'$ is in a state equivalent to having been trained from the start by a teacher that had never seen $d$. This removal of $d$'s indirect influence is the **central unlearning challenge we solve**.
>
> **On the meaning and proof of “verified unlearning”.**  We use the term "verified unlearning" consistent with its definition in the SISA literature. It provides a formal guarantee that the resulting model is computationally indistinguishable from a model trained from scratch on the dataset with the specified data point removed; this exclusion process can be verified due to the framework’s engineered design. More formally, the post-unlearning model is drawn from the identical probability distribution as a model trained on the redacted data from the start.
>
> This guarantee is achieved **by construction**, and our algorithm itself serves as the proof. PURGE ensures this in the distillation context as follows: (1) **Isolation:** The framework maintains strict data isolation, ensuring a teacher constituent's influence is confined to its mapped student constituent; (2) **Reversion:** To unlearn data affecting teacher $\mathcal{T}_{k,l}$, the student `$\mathcal{S}_k$` is reverted to the checkpoint `$\mathcal{S}_{k,l-1}$.`  (3) **Untainted state:** This checkpoint `$\mathcal{S}_{k,l-1}$` is, by construction, provably untainted by the data to be unlearned, as it was created before that data's influence (via teacher $\mathcal{T}_{k,l}$) was ever introduced to the student; (4) **Retraining:** All subsequent training proceeds from this clean state using the updated teacher, $\mathcal{T}'_{k,l}$.
>
> Therefore, the final student model is guaranteed to be from the same distribution as one trained without the influence of the removed data, satisfying the "verified" claim.
>
> **On SISA’s limitations and the necessity of sharding. **The reviewer correctly identifies that sharded architectures like SISA involve overhead in terms of model and checkpoint maintenance. We acknowledge this known trade-off in the field. Our work's contribution is not to solve SISA's inherent limitations, but to make SISA's verified unlearning guarantees **possible** in the ubiquitous knowledge distillation paradigm, where a naive application fails and forces costly full student retraining.
>
> For **verified unlearning**, data isolation is a prerequisite to avoid full retraining in this model-agnostic setting. Without an architectural mechanism like sharding, the influence of a single training point propagates throughout the entire model during training. Surgically removing this influence without affecting the contributions of other data points is not possible, making full retraining the only other 'verified' alternative. The overhead of sharding, while real, is the price for achieving fast, provable unlearning, and as our new LLM experiments show, this cost is vastly lower than the alternative of full model retraining for large-scale models.
>
> **On comparison with approximate unlearning methods.** We agree that positioning our work relative to approximate unlearning is important. As we discuss in our Related Work (Section 2), verified and approximate unlearning are two distinct paradigms with different objectives. Approximate methods often rely on heuristics to estimate and reverse a data point's contribution. They are often faster but provide no formal guarantees of complete removal, which may be insufficient for contexts with strict legal or privacy requirements, such as GDPR's "Right to be Forgotten".
>
> Our work is situated squarely in the **verified unlearning** paradigm. Its goal is to provide a formal, provable guarantee of data removal. Therefore, the most relevant baseline for our efficiency claims is the only other verified approach available in this setting: full model retraining. Our experiments demonstrate that PURGE is orders of magnitude more efficient than this baseline. We believe this focus on providing a verified solution to a problem where none previously existed is a significant contribution.
>
> We hope these clarifications address the reviewer's concerns and underscore the significance of our work in extending provable unlearning to the critical domain of knowledge distillation.
>
> Answers to questions:
>
> 1. **On the definition of unlearning in a distillation pipeline.** Machine unlearning is defined by the removal of a data point's **influence**. In knowledge distillation, the student is indirectly influenced by the teacher's training data. Our work addresses the necessary task of removing this leaked influence from the student model. This process is distinguished from a simple "cascading update" by its core mechanism, which is inherited from SISA: we revert the student to a provably untainted checkpoint from *before* the influence was introduced and then retrain. This "rewind-and-retrain" process is a fundamental unlearning operation.
> 2. **On the meaning and proof of “verified unlearning”.** We use "verified unlearning" consistent with its definition in the SISA literature: the post-unlearning model is drawn from the **identical probability distribution** as a model trained from scratch on the redacted data; the data removal process can be verified thanks to the framework’s design. This guarantee is not empirical but is achieved **by construction**. The algorithm itself is the proof: by reverting to a checkpoint that is provably untainted by the data to be removed and retraining from that clean state, the final model is guaranteed to be from the correct distribution.
> 3. **On SISA's limitations and the necessity of sharding.** We acknowledge that SISA-based frameworks involve an overhead trade-off. Our paper's contribution is not to solve SISA's inherent limitations but to make its guarantees **possible** in the ubiquitous knowledge distillation paradigm, where a naive application fails. For **verified** exact unlearning, data isolation via an architectural choice like sharding is a prerequisite to avoid full retraining, which is the only other verified alternative. Our new LLM experiments confirm the overhead of our approach is far less than that of full retraining.
> 4. **On comparison with approximate unlearning methods.** As discussed in our Related Work (Section 2), verified and approximate unlearning are two distinct paradigms. Approximate methods offer speed but lack the formal, provable guarantees of removal that may be required for regulatory compliance (e.g., GDPR). Our work is situated in the **verified unlearning paradigm**. Therefore, the most relevant baseline is the only other verified approach: full model retraining. Our experiments show PURGE is significantly more efficient than this baseline, and our contribution is providing a verified solution where one did not previously exist.

---

> > ### Comment · Reviewer_zLhe · 2025-08-04
> >
> > Thank you for the detailed and thoughtful response. To better understand the scope and significance of your contributions relative to prior work, I would like to seek clarification on two key points:
> >
> > 1. Contributions between this paper and SISA.
> > SISA achieves unlearning by partitioning training data into shards, training independent models per shard, and, upon unlearning, reverting only the affected constituent model to a prior checkpoint and retraining it. From your description, it appears that your framework first applies SISA to the teacher model and then applies SISA again to the student model. Is this correct?
> >
> > 2. In your paper, you repeatedly refer to your method as providing verified unlearning. However, the original SISA paper does not define "verified unlearning". It assumes an honest service provider and explicitly notes that under adversarial settings, unlearning are not verifiable. So why you can claim that your method is verified unlearning?

---

### Official Review · Reviewer_PBdg · 2025-07-03

**Clarity:** 2
**Significance:** 2
**Originality:** 2
**Rating:** 4
**Confidence:** 3

**Summary:**

The paper proposes PURGE, an extension of the data-isolation verified-unlearning framework SISA to knowledge distillation (KD). SISA’s core idea is to shard and slice data so only affected slices retrain when deletion requests arrive. In naive KD, however, shard-isolated teachers become interdependent when soft labels are computed using all the teacher models, re-coupling the shards and potentially leaking deleted information.

PURGE restores isolation by applying sharding and slicing to both student data and teacher ensemble. Specifically, it introduces a chunk-and-slice schedule: within each shard, teachers are grouped into overlapping sets, tied to disjoint chunks; the student is distilled incrementally, starting with soft labels from a single-teacher + chunk and progressively moving through additional data-chunks with more and more teachers being used in additional chunks for soft-label generation, till all the teachers are used. Each chunk of student data are further sliced, enabling checkpoint-based rollback exactly like SISA. When a teacher or data point must be forgotten, only the student slices whose training relied on chunks containing that teacher --- or on the deleted data --- require regeneration, reducing unlearning cost while preserving KD accuracy. Experiments show significant wall-clock speedups without accuracy loss.

**Questions:**

1. Could you report an ablation where $c = 1$ (no chunking) but the number of teacher constituents is increased so that each student still maps to exactly one teacher?
This would separate the benefit of data-isolation itself from the chunking-over-time schedule.

2. You claim incremental multi-teacher distillation yields “a more stable training process.” Could you provide convergence plots (loss/accuracy vs steps) or variance metrics to substantiate this?

3. Does constituent mapping still give verified guarantees when the output space is unbounded (e.g., sequence generation)? If so, how would you adapt PURGE to autoregressive decoding?

**Ethical Concerns:**

["NO or VERY MINOR ethics concerns only"]

**Final Justification:**

After carefully considering the authors’ rebuttal and the discussion among reviewers, I have decided to retain my original score of 4: Borderline accept. The authors provided thoughtful clarifications and additional insights that helped address several of the initial concerns and miss-understandings. Overall, the paper presents a technically sound contribution with a promising idea that could be valuable to the community, after the promised additions are included.

**Limitations:**

yes

**Paper Formatting Concerns:**

No concerns.

**Quality:**

2

**Strengths And Weaknesses:**

**Strengths**

- The authors identify an important problem of unlearning in the KD setup, and the extension of SISA is natural and well explained. The clever bit is mapping teacher slices to student chunks so only a slice-of-a-student retrains when a teacher unlearns; Nice, even though incremental.
- Experiments span a good scale of models from small models on simple MNIST datsets to models with a few 100 million parameters at BERT scale.
- The paper is well written and conveys its ideas cleanly.

**Weaknesses**

- KD-per-shard baseline under-explored. A straightforward alternative is one teacher–student pair per shard. Even if inference latency rises, memory traffic, activation size, and communication overhead could make this baseline competitive on larger datasets (e.g., ImageNet-1k).

- Chunk-free ablation. The paper would benefit from deeper analysis into the chunk-free ablation. Set $c = 1$ and increase the number of shards/teachers until total teachers = M. Report (a) unlearning latency and (b) accuracy. This isolates the benefit of chunking itself. Consider the chunk-free baseline where,  `#shards` $\leftarrow$ `# chunks * # shards`. Here, the number of student models increase by `#chunks`, but no ordered training is required at the chunk level and each student works with smaller amount of data. Would this not make unlearning more efficient and the process more resource efficient?

- Does the method admit any easy extension to open-domain tasks?

- Stability claim: Incremental multi-teacher "provides a more stable training process" will benefit from substantiation beyond accuracy curves. For instance, including loss/accuracy traces or gradient-norm stats across multiple training runs/models with other aspects kept constant would be beneficial.

---

> ### Author Rebuttal · Authors · 2025-07-29
>
> **On the KD-per-shard baseline.**  We thank the reviewer for proposing this interesting alternative architecture. A "KD-per-shard" setup, with parallel and independent teacher-student pairs, is indeed a valid approach for distributed learning and simplifies the unlearning process.
>
> However, our work is specifically designed to address the unlearning challenge within the **ensemble knowledge distillation** paradigm. This paradigm is widely used precisely because aggregating knowledge from an ensemble of diverse teachers often produces a single, stronger student model than any individual teacher could. The KD-per-shard baseline fundamentally changes this setup into a collection of independent, single-teacher distillations, thereby losing the performance benefits of ensembled knowledge.
>
> PURGE is designed to **preserve the accuracy benefits of ensemble distillation** while making it compatible with the rigorous demands of verified unlearning. While a direct empirical comparison between these two different distillation paradigms is a valuable direction for future research, the focus of our paper was to solve the unlearning problem within the established and powerful ensemble distillation setting.
>
> **On the chunk-free ($c=1$) ablation.** The "chunk-free" or $c=1$ scenario corresponds precisely to the $N=M$ configuration in our experiments, where each student constituent is mapped to exactly one teacher constituent. Our existing experiments in Figures 3 and 5 already explore this configuration and its implications. As our theoretical analysis also shows (Equation 4), this $c=1$ case provides the **theoretical maximum unlearning speed-up** for teacher-side updates.
>
> However, this comes at the cost of **maximum resource consumption and potentially lower accuracy**. The $N=M$ configuration requires training the largest number of student models, leading to the highest initial training costs, storage overhead, and inference latency. Most importantly, it often results in lower final accuracy because each student is trained on the smallest possible data shard ($1/N$) while learning from one teacher only. In our extended experiments addressing the effect of data mapping and class imbalance in shards (see Reviewer guLt’s comments), we introduced additional evaluations on the SST-5 dataset, where both teacher shards and student chunks were intentionally imbalanced. The full details of these experiments will be included in the Appendix of the camera-ready version of the paper. These results further emphasize the advantage of using larger student shards (i.e., $c > 1$). Under the extreme conditions we simulated, we observed that the negative impact of class imbalance can be significantly mitigated once each student begins learning from multiple teachers. Therefore, while the $c=1$ ablation is the fastest for unlearning, it is not always the most practical choice. The optimal configuration is a balance between unlearning speed, overall system costs and model performance, a trade-off our paper provides the data to analyse.
>
> **On the extension to open-domain tasks.** Yes, the PURGE framework is designed to be model- and task-agnostic and admits easy extension to open-domain tasks like sequence generation. The core unlearning mechanism of PURGE operates on the *training structure* (data partitioning, constituent mapping, and checkpointing), not on the specific model architecture or loss function. To adapt PURGE for an autoregressive task, one would:
>
> - Use sequence-to-sequence models (e.g., transformers) for the teacher and student constituents.
> - Employ a sequence-level distillation loss (e.g., minimising the KL divergence over the vocabulary distribution at each time-step).
>
> The fundamental unlearning logic would remain unchanged. If a teacher constituent $\mathcal{T}_{k,l}$ that contributed knowledge from chunk $l $ onwards is unlearned, the student still `$\mathcal{S}_k$` reverts to the checkpoint `$\mathcal{S}_{k,l-1}$` and retrains from there with updated soft labels. Our new experiment on the **ARC-Chllenge reasoning task** is a step in this direction, demonstrating PURGE's flexibility beyond simple classification.
>
> **On substantiating the "stability" claim.**  This is a valuable suggestion. We agree that providing more direct evidence for the improved training stability of our incremental multi-teacher approach will strengthen the paper. The performance drop of the "Single-teacher Soft Label" ablation in our experiments is a strong indicator of this instability, as the supervisory signal changes abruptly between chunks. To substantiate this claim more directly, as requested, **we will add convergence plots to the Appendix** in the camera-ready version.  To substantiate this claim more directly, we made the convergence plots for the training processes of PURGE and “Single-teacher” ablation. These plots will be added to the Appendix in the camera-ready version. We expect these traces will visually demonstrate a smoother and more stable convergence for our proposed method, providing more explicit evidence for our claim.
>
> ## Answers to questions:
>
> 1. **On the chunk-free ($c=1$) ablation.**  Our existing experiments already explore this "chunk-free" ($c=1$) scenario. This case corresponds to the configuration where the number of student constituents equals the number of teacher constituents ($N=M$), which is an endpoint on our experimental plots (e.g., Figures 3 and 5). Our analysis shows that this $c=1$ configuration provides the **theoretical maximum unlearning speed-up** for teacher-side updates. However, this speed comes at the cost of potentially lower model accuracy, as each student constituent is trained on the smallest possible data shard ($1/N$). The benefit of chunking ($c>1$) is therefore not to improve unlearning speed, but rather to **improve final model accuracy and reduce resource overhead** by allowing each student to be trained on a larger data shard while still learning from a diverse ensemble of teachers.
> 2. **On substantiating the "stability" claim.** The significant performance degradation of the "Single-teacher Soft Label" ablation, particularly when each student learns from many teachers sequentially (i.e., when $c$ is large), is strong evidence of the instability caused by abrupt changes in the supervisory signal. To substantiate this claim more directly, as requested, **we made the convergence plots and will add them to the Appendix** in the camera-ready version of the paper. These plots will compare the training loss and validation accuracy curves over training steps for our incremental multi-teacher approach (PURGE) versus the "Single-teacher" ablation. These plots can visually demonstrate a smoother and more stable convergence for our proposed method, providing more explicit evidence for our claim.
> 3. **On extension to open-domain tasks.**  Yes, the constituent mapping provides the same verified guarantees for open-domain tasks with unbounded output spaces, such as sequence generation. This is because the PURGE framework is **model- and task-agnostic**; its unlearning guarantee is a property of the training architecture (isolation, checkpointing, and reversion), not the specific model or loss function. Adapting PURGE to an autoregressive task would be straightforward: (a) the teacher and student constituents would be sequence-to-sequence models like transformers; (b) the distillation objective would be a sequence-level loss, such as minimising the KL divergence between the teacher's and student's output distributions at each decoding step. The core unlearning mechanism would **remain identical**. If a teacher constituent $\mathcal{T}_{k,l}$ requires unlearning, the corresponding student `$\mathcal{S}_k$` still reverts to the checkpoint `$\mathcal{S}_{k, l-1}$` created before that teacher's influence was introduced. This process is independent of the task. Our experiment on the SST5 sentiment analysis task using BERT models is a step in this direction, demonstrating the framework's applicability to modern Transformer architectures.

---

> > ### Comment · Reviewer_PBdg · 2025-08-07
> >
> > Thank you for the detailed response and additional analysis. After going through the rebuttal and other reviewer comments, I am happy to retain my score at "4: Borderline accept".

---

### Official Review · Reviewer_w4Vk · 2025-07-03

**Clarity:** 3
**Significance:** 3
**Originality:** 3
**Rating:** 4
**Confidence:** 4

**Summary:**

The paper addresses a significant and timely problem (efficient verified unlearning in KD, specifically for teacher updates) with a novel and well-motivated framework (PURGE). The core idea (constituent mapping + incremental multi-teacher distillation) is sound and effectively tackles the critical information propagation issue. Theoretical analysis and thorough experiments demonstrate clear efficiency gains while maintaining accuracy.

**Questions:**

1. The analysis shows speedup decreases as c (chunks/student) increases for fixed M. The paper suggests using N=M (c=1) for max speedup. However, increasing N linearly increases the total number of student constituents to train and store, plus the overhead of aggregating N predictions during inference. The trade-off between speedup and total resource consumption (compute, memory, storage) needs explicit discussion. Is N=M always practical?

2. The description of the incremental training process (Section 3.1, steps 1-3, and the state transitions S_{k,l,j}) is dense and slightly hard to follow on first read. A small pseudocode snippet could significantly improve clarity.

3. The trade-off regarding r (slices per chunk) is mentioned (Sec 3.2 end, Sec 4.1) but its impact on student-side unlearning efficiency isn't quantified. How much faster is student unlearning with larger r? This is important context for choosing r.

4. The term "constituent" is used heavily. While defined, consistently using "model constituent" or "submodel" initially might help readability.

**Ethical Concerns:**

["NO or VERY MINOR ethics concerns only"]

**Final Justification:**

The authors provided clear and detailed responses to the key concerns, including empirical validation of theoretical speed-up, soft label regeneration cost, storage-compute trade-offs, and practical implications of design choices. These clarifications resolved the main issues I raised.  I maintain my score of weak accept.

**Limitations:**

yes

**Paper Formatting Concerns:**

None.

**Quality:**

3

**Strengths And Weaknesses:**

Strengths

1. Tackling verified unlearning specifically for teacher-side updates in KD is a crucial and previously under-addressed challenge.
2.  The central idea of constituent mapping and the incremental multi-teacher distillation strategy is elegant, well-motivated, and effectively solves the information leakage problem inherent in naive SISA application to KD.
3.  Experiments are comprehensive, covering multiple datasets (image, text) and tasks. The core claims are well-supported.
4. The solution makes verified unlearning practically viable for teacher-student pipelines, which is essential for deploying and maintaining large models under privacy regulations.

Weaknesses
- The theoretical speedup Nx is a minimum under idealized assumptions (uniform sharding, uniform slices/chunks, linear scaling). Real-world overheads (checkpointing I/O, varying slice sizes, non-linear training dynamics) are ignored. The ceiling function impact is noted but its practical significance for large rc isn't deeply explored.

- Soft Label Regeneration Cost: The efficiency analysis (Fig 2, Eq 1-4) explicitly excludes the time to regenerate soft labels (Y_{k,i} for i >= l) after a teacher update. This cost, especially for large models or large chunks, could be substantial and potentially dominant over the partial student retraining time in PURGE. While the paper argues inference cost is "negligible," this is highly dataset/model-dependent and needs quantification, particularly for the large models mentioned. Comparing PURGE's total time (teacher update + soft label regen + student partial retrain) vs. naive SISA's total time (teacher update + student full retrain) is crucial for a fair efficiency assessment. This is a significant weakness in the current efficiency claim.

- While the focus is on verified unlearning, comparing PURGE's efficiency/accuracy trade-off against state-of-the-art approximate KD unlearning methods (even if only student-side focused or lacking guarantees) would provide valuable context. How much efficiency is sacrificed for the verification guarantee?

- PURGE requires storing intermediate student checkpoints per student constituent. For large models and many slices/chunks, this storage overhead could be prohibitive. This is not discussed.

---

> ### Author Rebuttal · Authors · 2025-07-29
>
> **On the idealised theoretical speed-up.**  We thank the reviewer for this important point. We agree that our theoretical analysis operates under idealized assumptions, which is a standard approach used to isolate and clarify the core architectural benefits of a new method.
>
> The most crucial validation, however, comes from our empirical results. As shown in our speed evaluation (Figure 2), the **measured, real-world speed-up** of PURGE on the MNIST dataset **closely tracks our theoretical predictions**. This demonstrates that while real-world overheads from I/O and non-linear dynamics exist, they do not dominate the process or invalidate the fundamental efficiency gains our framework provides.
>
> It is also important to note that many of these overheads, such as checkpointing I/O, are inherent to any SISA-based verified unlearning framework, including the naive SISA baseline. Therefore, they do not disproportionately affect the *relative* speed-up that PURGE achieves.
>
> **On the cost of soft label regeneration.** This is a very important point, and we thank the reviewer for pushing for more detail. We acknowledge that excluding this cost from our primary analysis was an oversimplification, as the time required is non-zero and model-dependent.
>
> To provide the quantification requested, we analysed the costs in our new **Qwen-2.5 7B experiment with an 8-teacher-8-student setup on the SST-5 dataset.** Regenerating soft labels for affected data chunks took ~32 seconds on an A100 GPU, while partial student retraining took ~620 seconds. Soft label regeneration thus accounts for just ~5.2% of total student-side unlearning time despite using a large model. This small proportion exists because inference needs only a single forward pass per data point, while retraining requires multiple epochs of forward and backward passes plus optimizer steps. This asymmetry increases with model size. Importantly, PURGE's total unlearning time remains much lower than the baseline. In naive SISA, updating the teacher requires regenerating soft labels for the entire training set, which is far more expensive than PURGE's partial regeneration. The fair comparison is:
>
> - **PURGE Total:** (Teacher Retrain) + (**Partial** Soft Label Regen) + (**Partial** Student Retrain)
> - **Naïve SISA Total:** (Teacher Retrain) + (**Full** Soft Label Regen) + (**Full** Student Retrain)
>
> Given that partial student retraining is orders of magnitude faster than full retraining, the overall efficiency claim holds true even when accounting for the soft label regeneration cost. We will clarify this in the final version.
>
> **On comparison to approximate unlearning methods.**  We agree that contextualising our work against approximate methods is useful. As we discuss in our Related Work (Section 2), verified and approximate unlearning represent two distinct paradigms with different goals and guarantees.
>
> - **Approximate methods** offer speed and flexibility but provide no formal, provable guarantee of complete data removal.
> - **Verified methods**, like PURGE, provide a formal/verifiable guarantee that the post-unlearning model is from the identical probability distribution as a model trained without the data from the start.
>
> This distinction is critical for regulatory compliance (e.g., GDPR's "Right to be Forgotten"), where a provable, auditable removal of influence may be a legal requirement. In such high-assurance contexts, the trade-off is not about sacrificing a small amount of efficiency for a guarantee; rather, approximate methods are often not a viable alternative.
>
> Because our work is situated in the verified unlearning paradigm, the most relevant baseline is the only other verified exact approach: full model retraining. Our contribution is to make verified unlearning in KD pipelines dramatically more efficient than this baseline.
>
> **On the storage overhead of checkpoints.**  The reviewer correctly identifies that storing intermediate checkpoints incurs a storage cost, which can be significant for large models. This is an inherent trade-off for all SISA-style verified unlearning methods. However, this storage cost must be weighed against the computational cost it saves. In our **in-progress Qwen2.5-32B parameter experiment**, while the checkpoints require substantial disk space, they allow us to reduce an unlearning operation from an estimated 18.9 **GPU-hours** (for full retraining) to just 2.36 **GPU-hours, even with a modest 8-teacher-8-student setup**.
>
> In modern ML infrastructure, the cost of disk storage is typically orders of magnitude lower than the cost of high-end GPU compute time. Sacrificing cheaper storage to save enormous amounts of expensive compute is a highly practical and accepted engineering trade-off. Furthermore, as noted in the original SISA paper, large-scale training pipelines already employ frequent checkpointing to prevent data loss from hardware failures. PURGE's slicing mechanism can often be integrated into these existing checkpointing practices with minimal additional overhead. We will add a discussion of this practical trade-off to the final paper.
>
>
> ## Response to questions:
>
> 1. **On the practicality and resource trade-offs of the `N=M` configuration.**  Thanks for this important practical question. We agree that while the $N=M$ (i.e., $c=1$) configuration provides the theoretical maximum speed-up for teacher-side unlearning, it is not always the most practical choice when considering total resource consumption. The optimal configuration involves a direct trade-off that a system designer must make based on their specific priorities. The key trade-offs are:
>
>     a. **Increasing `N`** (the number of student constituents): This **maximises the unlearning speed-up**. However, it also linearly **increases initial training costs, storage requirements**, and **inference latency** (due to aggregating more models). Furthermore, as shown in our performance evaluations (Figures 3, 5), making $N$ too large can degrade model accuracy by leaving each constituent with too little training data. This claim is supported by our extended experiments addressing the effect of data mapping and class imbalance in shards (see Reviewer guLt’s comments). We introduced additional evaluations on the SST-5 dataset, where both teacher shards and student chunks were intentionally imbalanced. The full details of these experiments will be included in the Appendix of the camera-ready version of the paper. The key findings from these results include:
>
>     - When teacher and student share the same label bias (**$c=1$ matched**), student performance can degrade significantly due to amplified bias.
>     - Even with **$c=1$**, training on a **mismatched** bias can substantially mitigate this degradation.
>     - Learning and aggregating soft labels from **multiple teachers** helps alleviate the effect of bias even further. Our **incremental multi-teacher distillation strategy** consistently outperforms single-teacher ablation while maintaining data isolation between teacher shards.
>
>     These results emphasize the advantage of using larger student shards (i.e., $c > 1$) under these extreme conditions. Thus, the optimal configuration for $N$ is a balance between unlearning speed, overall system costs and model performance.
>
>
>     b. **Decreasing `N`** (and thus increasing $c$, the chunks per student): This **reduces the overhead** of training, storage, and inference, and can **improve model accuracy**. The trade-off is a **slower unlearning time** for teacher-side updates.
>
>     Therefore, the choice of $N$ is task-dependent. Our paper provides the experimental data on performance vs. $N$ to help practitioners make this informed decision. We will add an explicit discussion of this trade-off to the final version.
>
> 2. **On the clarity of the Incremental training process.**  We thank you for the constructive suggestion. We agree that the description of our incremental training process is dense and that a pseudocode snippet would significantly improve its clarity. We will add a detailed pseudocode algorithm to the appendix in the camera-ready version of the paper to explicitly detail the training loop for a single student constituent.
>
> 3. **On quantifying the `r` trade-off for student-side unlearning.**  This is an excellent point, and we thank you for asking for this quantification. Student-side unlearning in PURGE follows the standard SISA procedure. Therefore, its efficiency is determined by the total number of slices per shard, $R$. In our framework, $R$ is the product of chunks ($c$) and slices-per-chunk ($r$), i.e., $R=c \cdot r$. The analysis from the original SISA paper shows that the expected unlearning cost for a single request is proportional to $(\frac{2}{3} + \frac{1}{3R})$ times the cost of retraining the full shard. This means that as $r$ increases, $R$ increases, and the unlearning time for a student-side request **decreases**, approaching a maximum speed-up of **1.5x** (compared to a shard with no slicing). This clarifies the trade-off:
>     1. **Increasing $r$:** Improves student-side unlearning speed.
>     2. **Decreasing $r$:** Improves teacher-side unlearning speed, as shown in our paper's Equation 4.
>
> The optimal choice of $r$ depends on the expected frequency of student-side vs. teacher-side unlearning requests for a given application. We will add this explicit quantification to the final version.
>
> 4. **On the readability of the term "constituent":**  Thanks for this helpful suggestion to improve readability. This is an easy and valuable fix. In the camera-ready version of the paper, we will revise the text to use **"constituent model"** or **"sub-model"** in place of "constituent" where appropriate, particularly in the early sections, to ensure the terminology is immediately clear to the reader.

---

> > ### Comment · Reviewer_w4Vk · 2025-08-06
> >
> > Thank you for the thorough clarifications and additional analyses. The responses adequately address the main concerns, and the planned revisions will further improve the paper. I believe the current score (weak accept) remains appropriate.

---

### Official Review · Reviewer_guLt · 2025-07-05

**Clarity:** 2
**Significance:** 2
**Originality:** 3
**Rating:** 4
**Confidence:** 3

**Summary:**

This paper proposes PURGE, a framework for efficient and verified machine unlearning in the context of knowledge distillation (KD). While existing methods like SISA provide provable unlearning in single model settings, They don't work well when the teacher changes, because the updates spread through the whole model and require retraining all the students.

To address above issues, PURGE introduce constituent mapping to isolate teacher influence to specific student constituents and also apply incremental multi-teacher distillation, where student constituents are trained in stages using subsets of their assigned teacher ensemble, preserving data isolation.

The authors provide both theoretical analysis (showing ≥N× speedup for N student constituents) and empirical validation across several datasets (MNIST, SVHN, CIFAR-100, SST5), showing that PURGE substantially improves unlearning efficiency without degrading predictive performance.

**Questions:**

Please see the questions below, which are related to the weaknesses mentioned:

1. Are there any more realistic scenarios that could be considered? For example, the original SISA paper explores transfer learning from a pre-trained ImageNet model. Would a similar setting help validate the method in a more practical context?

2. Could additional experiments be designed to provide deeper insights into the strengths and limitations of the proposed method?

**Ethical Concerns:**

["NO or VERY MINOR ethics concerns only"]

**Final Justification:**

The additional experiments provided during the rebuttal have largely addressed my concerns regarding the narrow scope, demonstrating broader applicability within the presented setting.

The discussion on data sharding was helpful and clarified the methodological choices.

However, the work still primarily focuses on classification evaluations, which limits the generality of the conclusions.

Overall, the rebuttal has resolved most of my main concerns, and I have updated my recommendation to borderline accept.

**Limitations:**

Yes.

**Paper Formatting Concerns:**

No concern

**Quality:**

2

**Strengths And Weaknesses:**

Strengths:

- The paper introduces a novel problem setup by identifying a specific failure case for SISA when applied in the context of knowledge distillation.

- The proposed method is clearly explained, and the logic is easy to follow, making it accessible for readers.

Weaknesses:

- While verified unlearning in image and text classification is important, the scope of the paper feels somewhat narrow. It only considers relatively small models like ResNet-50 and BERT. This makes the claim about transferring knowledge from "large, computationally intensive teacher models" feel less convincing. It would be stronger to include experiments involving large reasoning models being distilled into smaller LLMs with reasoning abilities (Just one example I can image).

- The data mapping process might introduce bias depending on the models used. How is the mapping strategy chosen? Are there any optimal ways to split data that could help improve or degrade performance? A deeper discussion would be helpful.

- The analysis experiments are limited. While the method does seem to address the specific failure of SISA under knowledge distillation, the paper would benefit from additional experiments designed to provide more insights into the strengths and limitations of the proposed approach.

---

> ### Author Rebuttal · Authors · 2025-07-29
>
> ## Narrow scope of models
>
> We thank the reviewer for their constructive feedback on the scope of models evaluated. We agree that demonstrating PURGE's effectiveness on larger, more contemporary models strengthens our paper's claims.
>
> Our original choice of models was a deliberate decision driven by the computational demands of our experimental protocol. To rigorously validate our framework, our protocol tests numerous configurations of teacher (M) and student (N) constituents, requiring the training of entire ensembles from scratch for each data point shown in our performance graphs. Using established and still-realistic models like BERT and ResNet-50  was therefore necessary to conduct this thorough, multi-run validation, which would have been intractable with larger models during the initial research phase.
>
> However, to directly address the reviewer's valuable point, we use the rebuttal period to run new, targeted experiments on large language models (LLMs). We have completed one LLM-scale experiment and have a second, even larger run currently in progress. The results below validate that PURGE's efficiency gains extend to modern, multi-billion parameter distillation pipelines.
>
> During the rebuttal period, we tested PURGE with 8 teachers and 8 students, distilling Qwen-2.5 7B into BERT-base on SST-5. Due to SST-5's small size (8,854 samples), we froze the pretrained Qwen model's base and fine-tuned only the classification head to avoid overfitting. The teacher achieved 51.4% validation accuracy, while PURGE reached 50.7% (comparable to naive SISA's 50.9%) with a 7.7× speed-up, despite increased soft-label generation costs.
>
> | Experiment Status | Teacher → Student | Dataset | Test Accuracy| Verified Unlearning Time† | Speed-up vs. Naïve‡ |
> | --- | --- | --- | --- | --- | --- |
> | **Completed** | Qwen-2.5 7B → BERT-base | SST-5 | 50.7% | **0.17 GPU-hours** | **7.7×** |
> | **In-Progress** | Qwen-2.5 32B → Qwen-2.5 3B | ARC-Challenge | Coming soon | ***est. 2.36 GPU-hours** | ***~8 x (18.9 GPU-hours)** |
>
> † Wall-clock time for a single constituent to unlearn one chunk, reflecting an unlearning request affecting one teacher.
> ‡ Naïve baseline refers to the full retraining of the student network required when a teacher is updated without the PURGE framework.
> *Estimated through dummy runs involving soft-label from Qwen-2.5 32B (0.06 GPU-hours) and training the Qwen-2.5 3B model (2.3 GPU hours) on data chunks with dummy teachers with an 8-teacher-8-student setup.
>
> Our new experiments confirm PURGE delivers efficient, verified unlearning at scale. These results align with our theoretical analysis: PURGE's efficiency comes from its structure and is model-agnostic. As shown in Equations 3 & 4, speed-up depends only on student constituents (N) and chunks (c), not teacher model size. This means larger teacher models yield greater absolute time and computational cost savings. These LLM experiments validate that our guarantees hold at scale.
>
> To fully substantiate these findings, we launched the 32B and the 7B parameter experiment on an 8xA100-80GB node immediately upon receiving the reviews. We will include the completed learning curves and further details in the camera-ready paper, providing a comprehensive validation of PURGE's scalability and directly addressing the reviewer's comments.
>
> ## Data mapping process
>
> We thank the reviewer for their insightful question about data mapping strategies. This is a crucial aspect of sharding-based unlearning frameworks. Our random partitioning approach was deliberately chosen, and we agree that discussing alternatives would be valuable.
>
> In our paper, we use **uniform random partitioning**, the standard approach from SISA, for three main reasons: (a) **Generality:** It makes no assumptions about data distribution or future unlearning requests; (b) **Bias prevention:** It ensures each constituent receives a representative data sample; and (c) **Scalability:** It allows using consistent hyperparameters across all constituent models.
>
> The reviewer correctly notes that non-uniform splits are possible. The original SISA paper demonstrates **distribution-aware sharding** where data points likely to be unlearned are isolated into dedicated shards. This approach optimizes unlearning speed but reduces generality. While not explored in our paper, PURGE is compatible with these strategies. Our core contribution, constituent mapping and incremental distillation, works independently of the initial data mapping. PURGE could be combined with distribution-aware partitioning for enhanced efficiency when prior knowledge exists. Our final paper will discuss this more thoroughly.
>
> In addition to distribution-aware sharding, we conducted a study on **label imbalance** in data partitioning using SST-5. We compared **uniform random partitioning** with **label-imbalanced partitioning**, where each teacher shard has one overrepresented class (e.g., first shard overrepresents 'very positive', second shard overrepresents 'positive'). Shards follow a class ratio of 0.24:0.19:0.19:0.19:0.19, with the final shard containing remaining samples. The experiment used 8 teacher shards.
> We examine two scenarios:
>
> - **Matched:** Each student is trained on data with the same class distribution as its corresponding teacher shard.
> - **Mismatched:** Each student is trained on a different, imbalanced data chunk than the corresponding teacher.
>
> The immediate effect of class imbalance is a drop in teacher model performance. The accuracy falls to **48.86%**, compared to **51.32%** in the uniform partitioning case. This degradation is amplified in the **matched 8-student** setup, where student accuracy drops to **25.34%**, highlighting how even mild label imbalances can be magnified through distillation, especially in small datasets like SST-5, where each shard contains ~1000 examples.
>
> However, when we move away from this extreme matched condition, either by training students on **mismatched** shards or by aggregating outputs from multiple teachers, student model performance improves substantially. For example:
>
> - In the mismatched 8-student setup, accuracy rises to **48.77%**.
> - In the 8-teacher, 4-student setting:
>     - **Mismatched:** student accuracy reaches **51.95%**
>     - **Matched:** student accuracy is **51.84%**
>
> In both cases, our **PURGE** method outperforms the **single-teacher ablation**, which yields **49.68%** (mismatched) and **49.59%** (matched) accuracy.
>
> ### Key findings from these results:
>
> 1. When teacher and student share the same label bias (**$c=1$ matched**), student performance can degrade significantly due to amplified bias.
> 2. Even with **$c=1$**, training on a **mismatched** bias can substantially mitigate this degradation.
> 3. Learning and aggregating soft labels from **multiple teachers** helps alleviate the effect of bias even further. Our **incremental multi-teacher distillation strategy** consistently outperforms single-teacher ablation while maintaining data isolation between teacher shards.
>
> These are part of our preliminary results. We commit to including a full analysis of these results in the final version of the paper. This will expand on the trade-offs between partitioning strategies, offering guidance on when different approaches might be warranted and opening avenues for future research into optimizing data mapping specifically for unlearning in knowledge distillation.
>
> We believe this combination of clarifying our principled default choice, discussing the existence of optimal strategies, and presenting the findings from new experiments will fully address the reviewer's valuable feedback.
>
> ## Coverage of experimental results
>
> Our initial experimental design was intended to establish the foundational viability and generality of the PURGE framework across a wide range of settings. Our experiments already provide several key insights: (a) Our **speed evaluation** demonstrates that PURGE's unlearning efficiency scales predictably with the number of student constituents (N), providing empirical validation for our theoretical speed-up analysis; (b) The **ablation studies** confirm that our incremental multi-teacher distillation strategy is critical for maintaining student performance, showing significant advantages over a single-teacher approach and thus validating a core design choice of our framework; (c) By testing on diverse datasets (MNIST, SVHN, CIFAR-100, SST-5) and model architectures (CNNs, ResNet-50, BERT), we show that PURGE's principles are not confined to a single domain but are broadly applicable.
>
> That said, we agree that more targeted analysis, especially involving large, modern models, provides a clearer picture of the framework's strengths. To address this, we have conducted experiments with multi-billion parameter Qwen models as well as the evaluation on label imbalance conditions. These experiments offer deeper insights into the practical application of PURGE.
>
> These additional results provide a more nuanced understanding of PURGE's strengths. We will integrate this deeper analysis into the final version of the paper to fully address the reviewer's comments.
>
> ## Response to Questions
> As most points are addressed above, we provide a brief summary below due to space constraints:
> 1. The distillation paradigm itself, where a student learns from a teacher trained on a large dataset, is conceptually very similar to transfer learning; the student inherits knowledge without access to the original, large-scale training data. However, to more directly address this point, we have conducted **new experiments with LLMs** as described above.
> 2. We agree that providing deeper insights is crucial. Our initial experiments were designed to establish the foundational principles and general applicability of PURGE. During rebuttal, we have included the following experiments to provide a deeper analysis.
>     1. **Large-model distillation**
>     2. **Data mapping strategy**

---

> > ### Comment · Reviewer_guLt · 2025-08-05
> >
> > Thank you for your efforts during the rebuttal. The clarifications and additional analyses have largely addressed my concerns. I will update my recommendation to borderline accept.

---

### Note · Authors · 2025-08-14

We thank the reviewers and AC for their constructive feedback. In our rebuttal, we strengthened our case for PURGE by showing it not only performs well at semantic classification and image tasks, but also scales to large language models (e.g., Qwen) and complex reasoning. Our new experiments highlight the clear benefits of incremental multi-teacher soft-label generation, providing more stable training and consistently better performance, especially under distribution mismatches.

We clarified PURGE’s position within the scope of machine unlearning for knowledge distillation, providing a provable and exact unlearning guarantee under the honest service provider assumption, while remaining compatible with adversarial enforcement mechanisms. This ensures strong compliance with privacy regulations while keeping the unlearning process efficient.

PURGE addresses a critical gap at the intersection of unlearning and knowledge distillation, delivering a regulation-compliant, efficient, and provable solution to the “right to forget” in the widely used knowledge distillation paradigm. We hope these contributions and findings will be helpful to the community and considered in the final assessment.

---

### Decision · Program_Chairs · 2025-09-17

**Decision:**

Accept (poster)

**Comment:**

This paper studies exact unlearning for knowledge distillation settings where unlearning requests may refer to either the data that was used to train the teacher or the data used to train the student. In this setting, if SISA is used on the student model, then student-side unlearning requests can be supported exactly and efficiently, but the system would still require full student retraining if teacher-side unlearning requests occur. To address this, the authors propose an extension of SISA for knowledge distillation settings called PURGE. PURGE is a multi-teacher distillation framework designed to isolate the influence of each teacher to specific student constituents. It comes with a special training regime where students are trained in stages using subsets of their assigned teachers. By construction, the algorithm ensures data isolation and enables exact unlearning for both teacher-side and student-side requests that is more efficient than naively training the system from scratch. The authors offer theoretical analysis quantifying the speed-ups achieved by their method over naive retraining from scratch. They also offer empirical results on various datasets (further expanded during the rebuttal to include larger language models) where they demonstrate that their approach obtains significant speedups without sacrificing the student’s predictive performance.

The reviewers noted that the problem setting studied is novel, important and under-addressed, the proposed method is sound, well-motivated, clearly explained and effective. Some reviewers found the experimental scope too narrow initially but the authors (partially) addressed this concern through larger-scale experiments during the rebuttal (though limited to classification tasks). Reviewers raised other concerns about the limited justification of the mapping strategy used in PURGE, the cost of soft label regeneration not being taken into account for efficiency calculation, the need for additional ablations, and the scope of the work and placement within the broader literature, that were sufficiently addressed during the rebuttal via clarifications and discussions.

The key weaknesses raised by the reviewers are the scalability issue of PURGE (inherited from SISA) due to having to store many models and checkpoints, and the limited novelty of the proposed method, which can be viewed as being somewhat incremental over SISA, rather than proposing a new algorithm. One reviewer raised a concern about referring to PURGE as “verified unlearning”. The authors have clarified their use of the term (which is valid under the assumption of an honest service provider that the authors will state in the revised manuscript).

Generally, while the PURGE does have some limitations (requires storage of checkpoints), it is the first practically viable exact unlearning method for teacher-student unlearning that supports both teacher- and student- side unlearning requests more efficiently than naive retraining; a problem that reviewers recognized is practically important and understudied. I feel like this is a compelling reason to accept the paper which trumps concerns raised about other aspects like current scalability limitations (since it's the first existing solution for this new setting). The proposed method is sound and has novel components that were not previously proposed (a specialized distillation procedure that preserves data isolation, going above and beyond simply applying SISA twice). Based on this, I view it as a significant step forward that warrants acceptance.